# The genetic architecture of age-related hearing impairment revealed by genome-wide association analysis

Erna V. Ivarsdottir [1,2], Hilma Holm [1], Stefania Benonisdottir[1], Thorhildur Olafsdottir[1], Gardar Sveinbjornsson[1], Gudmar Thorleifsson[1], Hannes P. Eggertsson [1], Gisli H. Halldorsson [1,2], Kristjan E. Hjorleifsson [1,3], Pall Melsted[1,2], Arnaldur Gylfason[1], Gudny A. Arnadottir [1], Asmundur Oddsson [1], Brynjar O. Jensson[1], Aslaug Jonasdottir[1], Adalbjorg Jonasdottir[1], Thorhildur Juliusdottir[1], Lilja Stefansdottir[1], Vinicius Tragante [1], Bjarni V. Halldorsson [1,4], Hannes Petersen[5,6], Gudmundur Thorgeirsson[1,5,7], Unnur Thorsteinsdottir[1,5], Patrick Sulem [1], Ingibjorg Hinriksdottir[8], Ingileif Jonsdottir [1,5,9], Daniel F. Gudbjartsson [1,2 ✉] & Kari Stefansson[1,5 ✉]

Age-related hearing impairment (ARHI) is the most common sensory disorder in older adults. We conducted a genome-wide association meta-analysis of 121,934 ARHI cases and 591,699 controls from Iceland and the UK. We identified 21 novel sequence variants, of which 13 are rare, under either additive or recessive models. Of special interest are a missense variant in *LOXHD1* (MAF = 1.96%) and a tandem duplication in *FBF1* covering 4 exons (MAF = 0.22%) associating with ARHI (OR = 3.7 for homozygotes, $P = 1.7 \times 10^{-22}$ and OR = 4.2 for heterozygotes, $P = 5.7 \times 10^{-27}$, respectively). We constructed an ARHI genetic risk score (GRS) using common variants and showed that a common variant GRS can identify individuals at risk comparable to carriers of rare high penetrance variants. Furthermore, we found that ARHI and tinnitus share genetic causes. This study sheds a new light on the genetic architecture of ARHI, through several rare variants in both Mendelian deafness genes and genes not previously linked to hearing.

[1] deCODE Genetics/Amgen, Reykjavik, Iceland. [2] School of Engineering and Natural Sciences, University of Iceland, Reykjavik, Iceland. [3] Department of Computing and Mathematical Sciences, California Institute of Technology, Pasadena, CA, USA. [4] School of Technology, Reykjavik University, Reykjavik, Iceland. [5] Faculty of Medicine, School of Health Sciences, University of Iceland, Reykjavik, Iceland. [6] Akureyri Hospital, Akureyri, Iceland. [7] Division of Cardiology, Department of Internal Medicine, Landspitali University Hospital, Reykjavik, Iceland. [8] National Institute of Hearing and Speech in Iceland, Reykjavik, Iceland. [9] Department of Immunology, Landspitali University Hospital, Reykjavik, Iceland. ✉email: daniel.gudbjartsson@decode.is; kari.stefansson@decode.is

Hearing impairment is a common sensory defect, affecting 1–2 out of every 1000 infants and over 50% of people over 80 years old[1,2]. Around 80% of prelingual hearing loss is caused by variants in the sequence of the genome[3], most commonly in the *GJB2* gene encoding the connexin 26 protein involved in inner-ear homeostasis[4]. Over 100 genes have been identified that cause prelingual or childhood-onset non-syndromic hearing loss, and 75% of those are inherited in a recessive manner (Hereditary Hearing Loss homepage, https://hereditaryhearingloss.org/).

Less is known about the genetics of age-related hearing impairment (ARHI), defined as a gradual decline of auditory function. ARHI is one of the most common chronic conditions affecting the elderly[5] and is associated with communication difficulties and reduced quality of life[6]. ARHI is usually caused by degeneration of the hair cells in the cochlea. The hair cells are specialized receptors that detect auditory stimuli and convert them into nerve signals that are transmitted to the brain[7]. ARHI can be treated, for instance with hearing aids or cochlear implants in severe cases. The heritability of ARHI has been estimated to be around 50% in twin studies[8]. The genetics of ARHI are complicated by the variability in onset, severity, and progression, as well as the effect of environmental factors such as noise exposure that can lead to hearing impairment[9]. Genome-wide association studies (GWAS) on ARHI have been performed[10–16] and a recent study based on UK Biobank self-reported hearing difficulty, reported 44 ARHI loci[17].

The standard type of hearing test is performed with an audiometer that delivers pure tones at different frequencies (measured in hertz (Hz)) and different intensities (measured in decibels hearing level (dB HL)). During the test, a sound is played at frequencies of 0.5, 1, 2, 4, 6, and 8 kHz, and each frequency at different intensity levels. The lowest intensity of sound detection for each individual is defined as their hearing threshold. According to the WHO classification of hearing loss, subjects with a hearing threshold above 25 dB HL are considered to have hearing impairment and the higher the thresholds the greater the impairment[18] (Supplementary Table 1). Hearing thresholds at frequencies 0.5, 1, 2, and 4 kHz were used in a pure tone average (PTA). These frequencies represent the range of speech.

Individuals with ARHI are at increased risk of tinnitus, the perception of a sound in the absence of an external sound. These phantom sounds are often described as ringing, buzzing, or hissing[19]. Most people experience tinnitus at some point in their life, but for 5–15% of the general population the tinnitus is incessant[20]. Treatment for tinnitus is lacking, even though 1–3% of individuals experience severe tinnitus affecting their life substantially, including difficulty with concentration and sleep[21]. A twin study estimated the heritability of tinnitus to be 56%[22], yet several genetic studies have failed to find associations of sequence variants with tinnitus[23].

To search for sequence variants associating with ARHI, we performed a GWAS meta-analysis of 121,934 cases and 591,699 controls from two non-overlapping Icelandic datasets and the UK Biobank (UKB). Subsequently, we assessed the effect of ARHI associating variants on tinnitus. Fifty-one independent sequence variants at 45 loci associate with ARHI, 41 under an additive model and 10 under a recessive model. Twenty-one of the associations have not been reported before, to the best of our knowledge. Using the association results, we furthermore constructed a GRS for ARHI.

## Results

### Summary of the data and demographics of ARHI in Iceland.
We conducted a GWAS of ARHI in three datasets obtained from the deCODE health study (DHS)[24], the National Institute of Hearing and Speech in Iceland (NIHSI) and the UKB (Fig. 1, Supplementary Table 2).

The DHS dataset is based on audiometric measures for 11,484 Icelanders, including 4140 ARHI cases (PTA > 25 dB HL) and 7344 controls, who are part of a comprehensive phenotyping of a general population sample enriched for carriers of rare and potentially high impact mutations[25]. The subjects were between 18 and 97 years of age at the time of recruitment (43.6% men; mean age = 55.4, standard deviation (SD) = 14.5, Supplementary Fig. 1a). The NIHSI is a clinic where patients are referred to for hearing and speech difficulties, and the NIHSI dataset consists of 36,905 audiometric measures of 22,212 Icelanders (55.5% men; mean age = 48.0, SD = 32.4, Supplementary Fig. 1b), of which 43.7% were performed on children (<18 years old). The NIHSI dataset is highly skewed toward those with ARHI, with a prevalence among adults of 73.6% for mild (PTA > 25 dB HL), 47.1% for moderate (PTA > 40 dB HL), 13.2% for severe (PTA > 60 dB HL) and 3.1% for profound (PTA > 80 dB HL) hearing impairment. Due to this bias, we defined the 9619 subjects with PTA above 25 dB HL as ARHI cases and designated 298,609 Icelanders with no available hearing data as population controls (excluding individuals in the DHS dataset). The UKB dataset consists of 108,175 cases with self-reported hearing difficulty and 285,746 controls of white British ancestry, at ages ranging between 40 and 69 years (45.6% men; mean age = 56.5, SD = 8.1).

The DHS dataset provides an opportunity to analyze the demographics of ARHI in Iceland, although we note that some individuals were recruited based on mutations causing or suspected to cause hearing impairment (Supplementary Table 3). The prevalence of hearing impairment was 36.1% for mild, 7.7% for moderate, 1.1% for severe and 0.1% for profound impairment. In line with previous studies[26,27], the prevalence of moderate hearing impairment at 75 years is 34% for men and 22% for women. The audiometric measures show that hearing declines with age at all frequencies but more drastically at the higher frequencies of 4–8 kHz (Table 1, Supplementary Fig. 2). The prevalence of mild hearing impairment reaches 5% shortly after 35 years and increases rapidly with age after 40; 18% at 50 years and 40% at 60 years (Supplementary Fig. 3). Consistent with previous reports[28], women are at greater risk of ARHI at low frequencies (0.5 and 1 kHz), while men are at more risk in the higher frequencies (≥2 kHz) (Table 1). Previous studies have observed an association between ARHI and short stature[29–31]. It has been postulated that the association is due to low levels of insulin-like growth factor 1 (IGF-1), which has a role in the development of the cochlea[29–31]. Performing a logistic regression on mild ARHI (>25 dB HL) against sex, age, and height, we observe that reduced height associates with increased risk of ARHI at the lower frequencies 0.5, 1, and 2 kHz (Table 1). After adjusting for height, the association with increased risk of ARHI in women at low frequencies (0.5 and 1 kHz) is no longer significant (Table 1). This indicates that the greater risk of ARHI at low frequencies for women is driven by the association with reduced height. These results replicate in the NIHSI dataset (Supplementary Table 4).

### GWAS meta-analysis.
To search for sequence variants associating with ARHI, we performed a meta-analysis of the three GWASs from DHS, NIHSI, and UKB, analyzing in total 46.9 million sequence variants under both additive and recessive models (Fig. 1). The UKB GWAS was performed on two imputed genotype datasets, one based on variants from the Haplotype Reference Consortium reference panel and the other based on

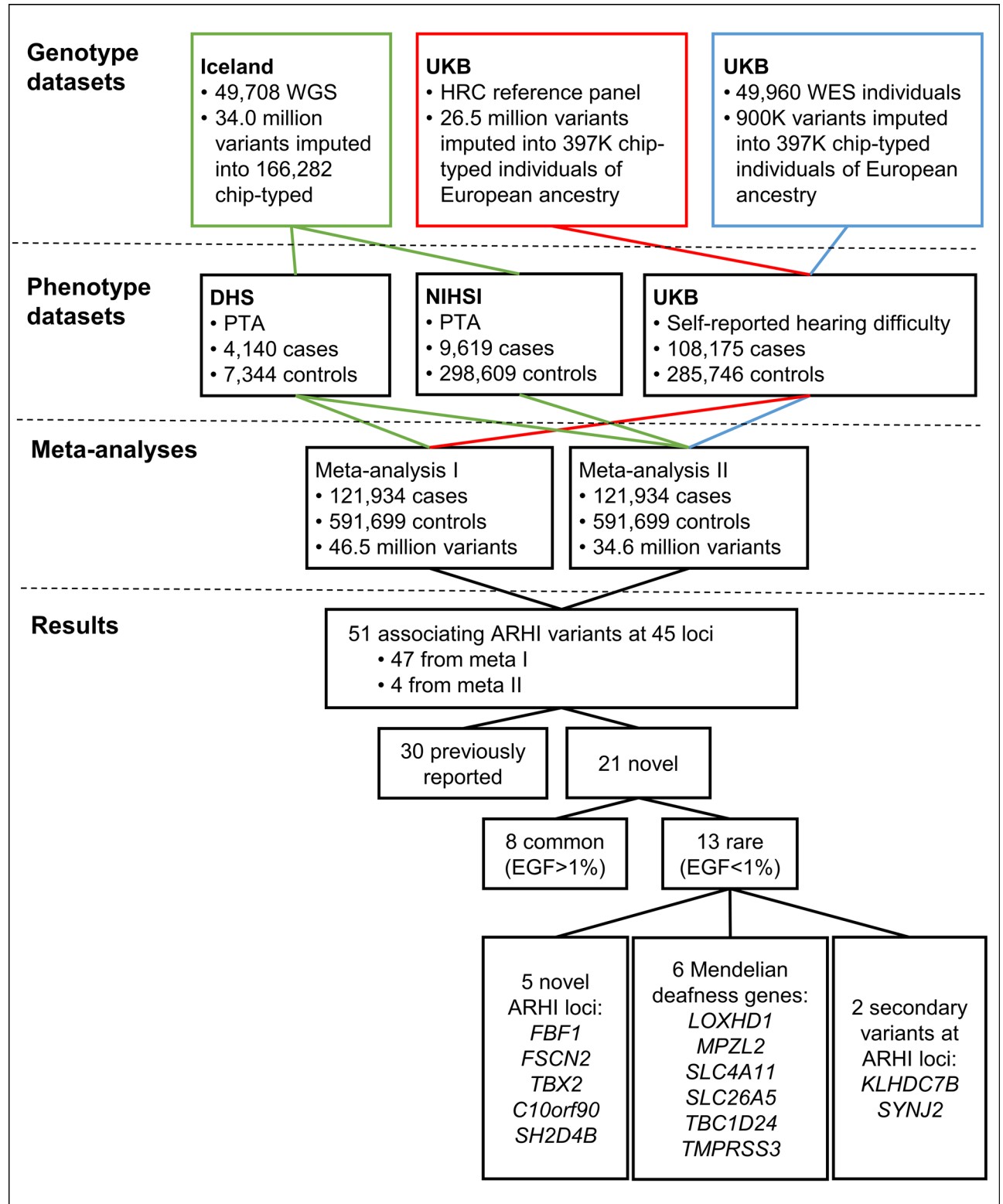

**Fig. 1 Study design and summary of results.** The UKB GWAS was performed with two genotype datasets marked in red and blue. The DHS and NIHSI GWAS was performed on one genotype dataset marked in green. WGS whole-genome sequenced, HRC Haplotype Reference Consortium, WES whole-exome sequenced, PTA pure tone average, DHS deCODE health study, NIHSI National Institute of Hearing and Speech in Iceland, EGF expected genotype frequency.

variants identified through whole-exome sequencing of 50 K study participants (see the "Methods" section). In total, 55 independent variants at 48 loci satisfied our genome-wide significance thresholds that are dependent on sequence variant annotation[32] ("Methods", Supplementary Table 5, Supplementary

Data 1 and 2, Supplementary Fig. 4). Because we do not restrict the definition of ARHI cases with respect to age at measure or severity, we might detect rare variants in the meta-analysis that are causing prelingual or childhood-onset hearing loss instead of ARHI. Due to this, we used the audiometric measures in the

**Table 1 Summary of audiometric measures from the DHS dataset ($N = 11,484$).**

| kHz | Mean | SD | Range | Hearing impairment prevalence (%) | | | | Effect of age and sex on ARHI | | | | Effect of age, sex, and height on ARHI | | | | | |
| | | | | Mild | Moderate | Severe | Profound | Age | | Sex | | Age | | Sex | | Height | |
| | | | | | | | | Effect | P-value | Effect | P-value | Effect | P-value | Effect | P-value | Effect | P-value |
| 0.5 | 22.1 | 6.3 | 20–100 | 11.0% | 2.5% | 0.5% | 0.1% | 2.90 | $6.1 \times 10^{-191}$ | 1.52 | $2.4 \times 10^{-10}$ | 2.76 | $1.2 \times 10^{-162}$ | 1.09 | 0.37 | 0.80 | $1.1 \times 10^{-6}$ |
| 1 | 23.2 | 7.7 | 20–100 | 15.6% | 4.3% | 0.7% | 0.1% | 2.80 | $5.4 \times 10^{-232}$ | 1.21 | $6.9 \times 10^{-4}$ | 2.70 | $1.3 \times 10^{-202}$ | 0.93 | 0.36 | 0.83 | $7.4 \times 10^{-6}$ |
| 2 | 26.1 | 11.1 | 20–100 | 27.8% | 9.5% | 2.2% | 0.3% | 3.50 | $<1 \times 10^{-300}$ | 0.78 | $2.6 \times 10^{-7}$ | 3.43 | $<1 \times 10^{-300}$ | 0.67 | $6.3 \times 10^{-9}$ | 0.90 | $1.7 \times 10^{-3}$ |
| 4 | 33.4 | 16.7 | 20–100 | 49.1% | 26.6% | 8.7% | 1.4% | 5.50 | $<1 \times 10^{-300}$ | 0.29 | $4.2 \times 10^{-139}$ | 5.44 | $<1 \times 10^{-300}$ | 0.27 | $1.9 \times 10^{-81}$ | 0.93 | 0.038 |
| 6 | 32.8 | 16.8 | 20–100 | 45.8% | 25.1% | 8.8% | 1.5% | 5.32 | $<1 \times 10^{-300}$ | 0.37 | $3.1 \times 10^{-96}$ | 5.28 | $<1 \times 10^{-300}$ | 0.35 | $1.3 \times 10^{-53}$ | 0.96 | 0.21 |
| 8 | 34.2 | 18.7 | 20–90 | 45.5% | 28.9% | 12.2% | 2.4% | 8.66 | $<1 \times 10^{-300}$ | 0.46 | $6.4 \times 10^{-52}$ | 8.60 | $<1 \times 10^{-300}$ | 0.43 | $3.5 \times 10^{-30}$ | 0.96 | 0.21 |

*kHz* kilohertz.
For each frequency, the mean, standard deviation (SD), and range of the hearing thresholds are shown. The prevalence of mild, moderate, severe and profound hearing impairment is shown. The effect (OR) of age in SD, sex given for women, and height in SD on ARHI (PTA > 25 dB HL) and the corresponding *P*-values, obtained using a likelihood-ratio test, are shown for each frequency.

Icelandic datasets to estimate the predicted hearing threshold of the carriers in childhood and observed that four of the 55 variants cause prelingual or childhood-onset hearing loss rather than ARHI ("Methods", Supplementary Data 2). Two are variants in *GJB2* known to cause deafness[33,34]. The other two variants, only detected in Iceland, are fully penetrant loss-of-function variants in known Mendelian deafness genes that have not been described before; a stop-gained variant in heterozygous state in *EYA4* (p.Tyr285Ter, MAF = 0.01%, OR = 35.6, $P = 1.1 \times 10^{-7}$; a likelihood-ratio test was performed in all logistic regression associations) and a frameshift variant in homozygous state in *OTOA* (p.Ala988ArgfsTer3, MAF = 0.65%, OR = 159.60, $P = 2.7 \times 10^{-20}$), where all carriers have moderate to profound hearing loss.

The UKB dataset has the largest sample size of the three datasets and the ARHI associations for the common variants are largely driven by the results from that dataset. Fourteen of the ARHI variants did not associate with ARHI in the Icelandic datasets ($P > 0.05$, Supplementary Data 3). However, the effects show a consistent direction in the three datasets and the pair-wise correlation coefficients between effect sizes are >0.56 (Supplementary Fig. 5). Thirty of the associations correspond to previously reported ARHI variants[14,17,35,36]. At 6 of those loci, the previously reported variants are non-coding[17], while we identified missense or splice region variants at these loci ($r^2 > 0.85$ between our top variant and the reported variant) (Supplementary Data 1). Thirteen of the associations not reported before, are represented by rare variants, of which six are located in Mendelian deafness genes and two are secondary associations at previously reported ARHI loci (Fig. 1). Through a gene-based burden test, where rare loss-of-function variants (MAF < 2%) in the same gene were aggregated and tested together, we identified one additional ARHI gene; *AP1M2*.

**Rare variants associating with risk of ARHI**. We found 16 ARHI variants that have rare genotypes with large effects (expected genotype frequency (EGF) < 1.0%), either in the heterozygous state ($N = 10$, MAF < 0.5%) or homozygous state ($N = 6$, MAF < 10.0%) (Table 2, Fig. 1). In Iceland, 4.9% of the population carries at least one of the 16 rare ARHI genotypes with large effects, and of those carriers that are older than 55, 72% have ARHI compared to 55% of non-carriers (DHS dataset). Overall, carriers of rare ARHI variants have a 2.2-fold (95% confidence interval (CI) = [1.8;2.7], $P = 1.0 \times 10^{-12}$) greater risk of mild hearing impairment than the rest of the population, 3.0-fold (CI = [2.1;4.3], $P = 1.4 \times 10^{-9}$) greater risk of moderate and 5.6-fold (CI = [3.1;10.2], $P = 1.9 \times 10^{-8}$) greater risk of severe impairment (DHS dataset, Fig. 2a).

Five of the variants that have rare genotypes with large effects are at loci that have not been reported for any type of hearing impairment: *FBF1*, *FSCN2*, *C10orf90*, *SH2D4B*, and *TBX2* (Supplementary Note 1).

A rare tandem duplication in *FBF1*, only detected in Iceland, associates strongly with ARHI (MAF$_{Ice}$ = 0.22%, OR = 4.2, $P = 5.7 \times 10^{-27}$). The variant is highly penetrant, with 81.5% of the 166 carriers having at least mild ARHI and 57.4% having moderate to profound hearing impairment (Fig. 3a). The duplication spans 7282 base pairs covering exons 4–7 of *FBF1*. To investigate the effects of the duplication on the transcription of the gene, we analyzed RNA sequencing data[37,38] from whole-blood of heterozygous carriers and non-carriers ($N = 13,067$) (Supplementary Note 2). Out of 60 heterozygous carriers, we found evidence of transcripts containing duplication of exons 4–7 defined by an extra splicing between exon 7 and 4 (Fig. 3b). This transcript isoform was not detected in RNA sequences from any

**Table 2 Association of sequence variants with ARHI.**

| Model | P-value | OR | rs name | Position | Chrom | EA | OA | Gene | Variant annotation | EAF ice (%) | EAF UK (%) |
|---|---|---|---|---|---|---|---|---|---|---|---|
| A | $5.7 \times 10^{-27}$ | 4.20 | – | 75927880 | 17 | T | C | FBF1* | CDS tdup | 0.22 | 0 |
| A | $8.0 \times 10^{-14}$ | 1.32 | rs146694394 | 158076728 | 6 | G | C | SYNJ2 | Missense | 0.37 | 0.46 |
| A | $1.8 \times 10^{-13}$ | 1.28 | rs141952919 | 103421378 | 7 | G | A | SLC26A5 | Missense | 0.33 | 0.52 |
| A | $5.9 \times 10^{-13}$ | 6.59 | rs761934676 | 2497068 | 16 | T | A | TBC1D24 | Missense | 0 | 0.01 |
| A | $5.1 \times 10^{-11}$ | 1.08 | rs113784020 | 118689022 | 11 | T | C | [PHLDB1] | Intergenic | 3.75 | 4.27 |
| A | $2.5 \times 10^{-10}$ | 1.92 | rs749405486 | 50549067 | 22 | A | AG | KLHDC7B | Frameshift | 0.00 | 0.06 |
| A | $4.9 \times 10^{-13}$ | 0.95 | rs72622588 | 182285702 | 3 | T | G | [FLJ46066] | Intergenic | 10.73 | 10.81 |
| A | $6.5 \times 10^{-10}$ | 1.20 | rs143796236 | 81528943 | 17 | T | C | FSCN2* | Missense | 0.21 | 0.74 |
| A | $7.6 \times 10^{-10}$ | 1.03 | rs13171669 | 149221680 | 5 | G | A | ABLIM3 | Intron | 42.92 | 42.34 |
| A | $1.2 \times 10^{-9}$ | 1.03 | rs3014246 | 45620405 | 1 | C | T | CCDC17 | Missense | 27.01 | 29.64 |
| A | $1.6 \times 10^{-9}$ | 1.03 | rs920701 | 75842965 | 13 | C | T | LMO7 | Intron | 34.91 | 36.67 |
| A | $3.9 \times 10^{-9}$ | 0.97 | rs11881070 | 2389142 | 19 | G | C | TMPRSS9 | upstream gene | 30.15 | 28.79 |
| A | $4.9 \times 10^{-8}$ | 1.81 | rs764272881 | 3228565 | 20 | T | A | SLC4A11 | Missense | 0.45 | 0 |
| A | $4.1 \times 10^{-8}$ | 71.17 | rs765488721 | 61403195 | 17 | T | C | TBX2* | Stop-gained | 0.01 | 0 |
| A | $8.3 \times 10^{-8}$ | 1.49 | rs727503493 | 42389042 | 21 | T | TG | TMPRSS3 | Frameshift | 0.22 | 0.07 |
| R | $1.7 \times 10^{-22}$ | 3.65 | rs118174674 | 46557437 | 18 | T | C | LOXHD1 | Missense | 2.95 | 1.99 |
| R | $3.2 \times 10^{-11}$ | 1.05 | rs9394952 | 43433367 | 6 | G | A | ABCC10 | Splice region | 49.99 | 48.13 |
| R | $7.7 \times 10^{-11}$ | 2.35 | rs12784122 | 80649861 | 10 | A | G | SH2D4B* | Downstream gene | 2.66 | 2.29 |
| R | $3.5 \times 10^{-10}$ | 17.32 | rs139123090 | 126459169 | 10 | G | G | C10orf90* | Missense | 0.89 | 0.47 |
| R | $3.2 \times 10^{-9}$ | 0.94 | rs557563970 | 117960109 | 1 | CGT | C | WDR3 | 3 prime UTR | 21.76 | 41.74 |
| R | $1.2 \times 10^{-8}$ | 4.79 | rs74543584 | 118262596 | 11 | A | T | MPZL2 | Missense | 1.47 | 0.83 |

OR odds ratio, Chrom chromosome, EA effect allele, OA other allele, EAF effect allele frequency, Ice Iceland, A additive model, R recessive model.The table lists the 21 variants revealed through the GWAS meta-analysis on ARHI. Gene names marked with * have not been linked to hearing, to the best of our knowledge. For intergenic variants, the nearest genes are reported in brackets. A likelihood-ratio test was performed in all genome-wide associations.

of the 13,007 non-carriers. We estimate that transcripts with splicing between exon 4 and 7 represent 7.5% (CI = [5.7;9.2]) of *FBF1* transcripts in carriers. Three other variants are correlated ($r^2 > 0.8$) with the duplication but none of them are coding (Supplementary Fig. 6a).

Six of the variants not reported before, with rare genotypes and large effects on ARHI, are coding variants located in Mendelian deafness genes: *LOXHD1*, *MPZL2*, *SLC4I1*, *SLC26A5*, *TBC1D24*, and *TMPRSS3*. Rare variants in these genes have been reported to cause severe to profound hearing impairment described as either prelingual or childhood-onset (DFNB77, DFNB111, DFNB61, DFNA65, DFNB86, and DFNB8; OMIM #613079, #618145, #217400, #613865, #616044, #614617, and #601072). However, apart from *TMPRSS3*, the ARHI associations we find in these genes are with missense variants that have milder effect than the prelingual or childhood-onset variants (Supplementary Note 3, Table 2, Fig. 4a–d).

The missense variant in *LOXHD1*, p.Arg1090Gln, on chromosome 18q21.1, associates with increased risk of ARHI under the recessive model (OR = 3.92, $P = 8.9 \times 10^{-22}$). P.Arg1090Gln has a MAF of 2.96% in Iceland and 1.99% in the UK and reaches genome-wide significance in both Iceland and the UK (Supplementary Data 1). No other variants are correlated with p. Arg1090Gln ($r^2 < 0.4$, Supplementary Fig. 6b). The variant has high penetrance, with 82.3% of the 62 homozygotes in the Icelandic datasets having ARHI and 48.4% having moderate to profound hearing impairment (Fig. 4a). We had information on hearing aid usage and the age when hearing aid usage started for 8211 of the 11,484 individuals in the DHS dataset and found that 15 out of 37 homozygous carriers use hearing aids and started using them earlier (mean age = 46.3 years, SD = 15.7) than heterozygotes and non-carriers (mean age = 60.0 years, SD = 14.8, P = 0.031; t-test). Heterozygotes are also at increased risk of ARHI (OR = 1.07, $P = 2.0 \times 10^{-4}$). However, this risk is much lower than the risk of homozygotes and therefore the effect of this variant deviates from the additive model ($P = 3.3 \times 10^{-12}$, Supplementary Data 4).

A rare frameshift variant, c.208delC, in *TMPRSS3* associates with ARHI under the additive model (OR = 1.49, $P = 8.3 \times 10^{-8}$, $MAF_{Ice} = 0.22\%$, $MAF_{UK} = 0.07\%$). In the homozygous state, c.208delC has been reported to cause congenital deafness (OMIM # 601072), but the increased risk of ARHI of heterozygous carriers has not been reported. In the Icelandic datasets, the only homozygous carrier has profound hearing loss. In the heterozygous state, the variant shows variable expressivity (Fig. 4e); where some carriers have normal hearing, 52 carriers have moderate, 17 have severe and 4 have profound hearing loss. Battelino et al. reported a Slovenian family-trio with congenital profound hearing loss, where the mother and the son were homozygous carriers of c.208delC and the father was a heterozygous carrier of c.208delC in *TMPRSS3* and c.35delG in *GJB2*[39]. Our results suggest that one copy of c.208delC in *TMPRSS3* can cause profound hearing loss.

**An ARHI gene detected with a burden test**. Using a loss-of-function variant gene-based burden test, the gene *AP1M2*, on chromosome 19p13.2, associates with ARHI under the recessive model in the Icelandic datasets (OR = 28.9, $P = 4.6 \times 10^{-7}$). Twelve homozygotes or compound-heterozygotes for loss-of-function variants with MAF < 2% in *AP1M2* had been invited to participate in the deCODE health study. Nine of them participated, three homozygous carriers of a stop-gained variant, p.Arg386Ter (MAF = 0.22%), three homozygous carriers for a splice donor variant, c.673 + 2T>C (MAF = 0.37%), and three compound-heterozygous carriers of these two variants. One

a)

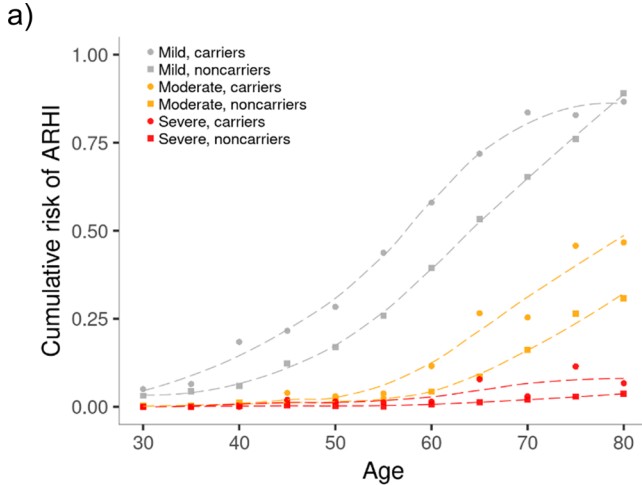

b)

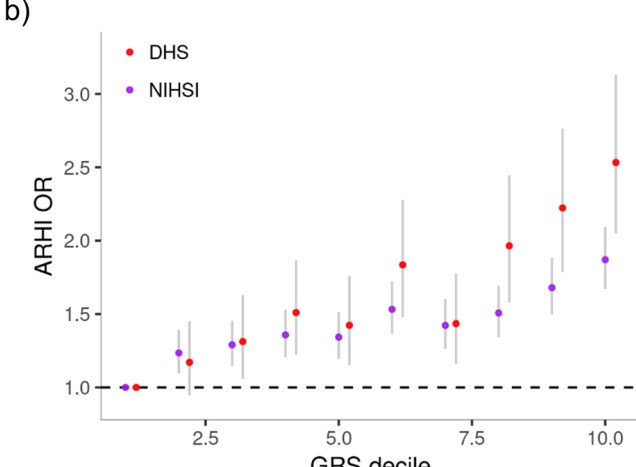

c)

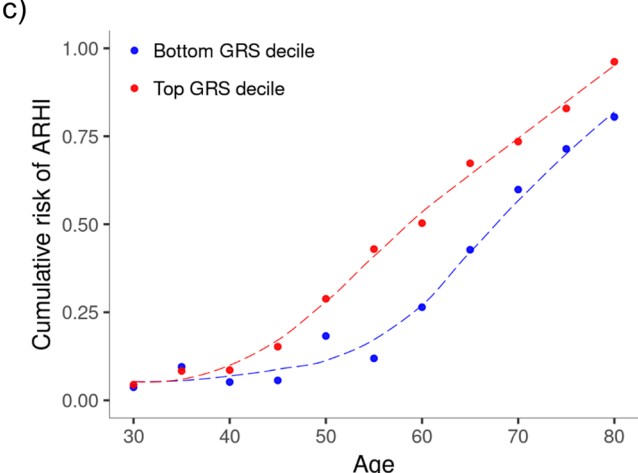

**Fig. 2 ARHI risk for rare variants and a common variant GRS. a** The fraction of individuals with mild (gray), moderate (orange), and severe (red) hearing impairment in the DHS dataset ($N = 11,484$) among the 4.9% of subjects that are carriers of any of the 16 rare ARHI variants (dots) and the 95.1% that are not carriers (squares), shown for different age groups. The corresponding dashed lines are polynomial regression lines of best fit. **b** The risk of ARHI for each GRS decile compared to the bottom decile expressed in ORs. The ORs from the DHS dataset (4140 cases and 7344 controls) are represented with red dots and the ORs from the NIHSI dataset (9619 cases and 298,609 controls) are represented with purple dots. The gray lines represent 95% confidence intervals. The dashed line is the reference for the bottom GRS decile (OR = 1). **c** The fraction of individuals with ARHI among subjects in the DHS dataset ($N = 11,484$) in the bottom GRS decile in blue and the top GRS decile in red, shown for different age groups. The corresponding dashed lines are regression lines of best fit.

hearing loss has been linked to chromosome 19p13.2 in families from Pakistan[43] and Germany[44], without a specific gene being implicated.

**The effect of the variants on ARHI by genotype**. It is interesting that 19% of the variants associate with ARHI under the recessive model, a much higher fraction than other age-related diseases. To further explore the effect of all of the ARHI associating variants per genotype we tested them under the genotypic model estimating the effect of heterozygous and homozygous carriers separately (Supplementary Data 4). We found that p.Arg402Gln in *TYR*, p.Val504Met in *KLHDC7B*, p.Thr656Met in *SYNJ2,* and p.Leu113Val in *CLRN2* have stronger effects on homozygotes than expected under the additive model ($P < 0.05$). Furthermore, we found that the variants in *ILDR1*, *CHMP4C*, and *CCDC68*, reported before as additive[17], are better explained by the recessive model, only showing significant effects on homozygous carriers (Supplementary Data 4).

**Dimensions of the audiometric data**. In the Icelandic datasets, the subjects underwent an audiometric test providing more information about the severity of the ARHI and the affected frequencies than in the UKB dataset. To further explore which hearing frequencies are affected by the ARHI variants, we tested each frequency (0.5, 1, 2, 4, 6, and 8 kHz) separately for association with the ARHI variants (Supplementary Data 3, Fig. 5). Most of the ARHI variants have similar effects at all frequencies although some variants have stronger effects on lower frequencies (Fig. 5b) and others on higher frequencies (Fig. 5a). For instance, p.Arg1090Gln in *LOXHD1* affects the lower frequencies more than higher frequencies under a recessive model, with the greatest effect on 1 kHz (OR = 8.4, $P = 1.6 \times 10^{-18}$), which is different from its effect on ARHI at 6 and 8 kHz ($P_{\text{het}} < 0.02$; Q-test). Furthermore, six ARHI variants, that do not associate with PTA-based ARHI in Iceland ($P > 0.05$), associate nominally with ARHI for some particular frequency (Supplementary Data 3, Fig. 5).

**Association of ARHI variants with tinnitus**. We tested the ARHI variants for association with tinnitus using self-reported information from DHS and UKB ($N_{\text{cases}} = 47,657$, $N_{\text{controls}} = 111,607$, Supplementary Table 7). ARHI variants detected under the additive model were tested for tinnitus using the additive model and ARHI variants detected under the recessive model were tested for tinnitus using the recessive model. Thirteen ARHI variants associate with tinnitus, controlling the false discovery rate at 0.05 using the Benjamini–Hochberg procedure (Fig. 6a; variants with lower OR shown in detail in Fig. 6b; Supplementary

additional compound-heterozygous carrier had audiometric measures from the NIHSI. Four of the individuals have severe, two have moderate and two have mild ARHI. Five of them reported the use of hearing aids, and their average age when starting using hearing aids (mean = 27.8, SD = 14.9) is substantially younger than that of other hearing aid users (mean = 60.2, SD = 15.1, $P = 0.022$; t-test). RNA and protein expression analyses of inner-ear tissue[40–42] have shown that Ap1m2 is expressed 7-fold higher in hair cells than in non-hair cells in mice with a false discovery rate of $3.4 \times 10^{-3}$, suggesting a specific role in hair cell function (Supplementary Table 6). Non-syndromic

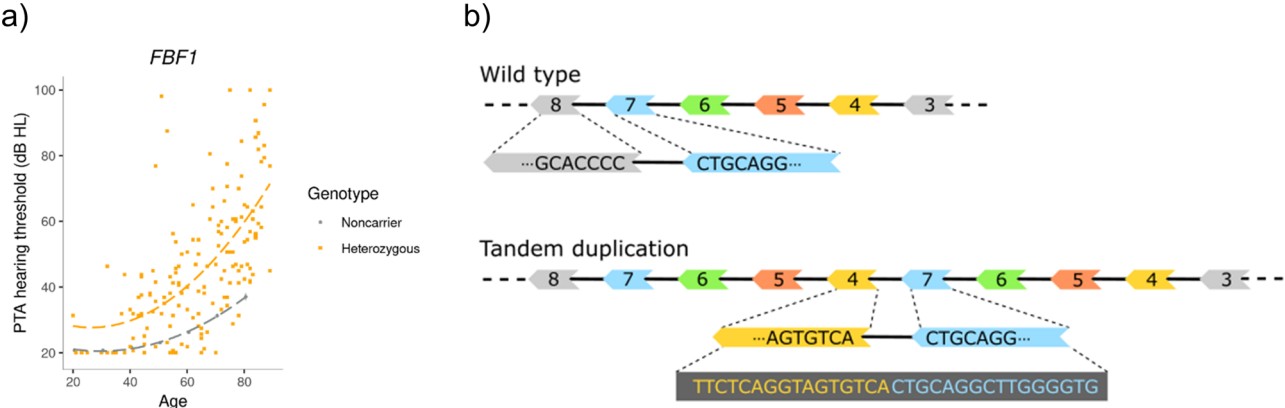

**Fig. 3 A tandem duplication in *FBF1* associates with ARHI. a**) The PTA hearing threshold of the heterozygous carriers (N=166) of the tandem duplication in *FBF1* in DHS and NIHSI datasets are indicated by yellow squares and the average PTA hearing thresholds of non-carriers in the DHS dataset ($N = 11,484$) are represented with gray dots. The corresponding dashed lines are polynomial regression lines of best fit. The gray vertical lines represent 95% confidence intervals. **b**) The exon structure of the wild type and the tandem duplication variant in transcript ENST00000586717.5 of *FBF1* on the reverse strand of chromosome 17. The exons of the transcripts are numbered, and solid black lines represent splicing between exons. The tandem duplication creates a longer transcript with extra sets of exons 4–7 that leads to novel splice junction starting at the end of exon 7 and splicing into the beginning of exon 4. The adjacent sequences of the exons are shown, and together they form the 32 bp sequence used for identification of the novel junction.

Table 7). Variants in *CTBP2, CRIP3, AGO2, PHLDB1, LMX1A, SLC26A5, ACADVL, SYNJ2,* and *CLRN2* associated with tinnitus under the additive model and variants in *ILDR1, ABCC10, SH2D4B,* and *C10orf90* associated with tinnitus under the recessive model. For all of the thirteen variants, the ARHI risk-increasing allele increases the risk of tinnitus, and the effect of all the ARHI variants on ARHI risk and tinnitus risk are highly correlated ($r = 0.72$, $P = 6.2 \times 10^{-8}$ and $r = 0.86$, $P = 6.0 \times 10^{-4}$ for the additive and recessive model, respectively; *t*-test, Fig. 6).

**Genetic risk score predicts ARHI risk**. We constructed a genetic risk score (GRS) for ARHI, based on the 35 ARHI variants with EGF > 1%, using effect sizes from the UKB dataset. The GRS associates with ARHI in both Icelandic datasets (OR = 1.31, CI = [1.25;1.37], $P = 4.1 \times 10^{-29}$ and OR = 1.18, CI = [1.15;1.21], $P = 7.5 \times 10^{-39}$ in DHS and NIHSI datasets, respectively) and the association is dose-dependent over GRS deciles (Fig. 2b). In the DHS dataset, individuals in the top decile of the GRS have 2.5-fold (CI = [2.0;3.1], $P = 6.1 \times 10^{-18}$) greater risk of ARHI than those in the bottom decile. Comparing the cumulative risk of ARHI against age between the top and bottom GRS deciles, shows that individuals in the bottom decile have their ARHI 10 years later than those in the top decile (Fig. 2c). Furthermore, individuals in the top GRS decile have a 3.2-fold (CI = [2.1;4.8], $P = 2.1 \times 10^{-8}$) and 2.7-fold (CI = [1.1;6.7], $P = 0.031$) greater risk of moderate and severe hearing impairment, respectively, than those in the bottom decile. If we compare the 4.9% who carry any of the 16 rare ARHI variants to the bottom 10% of the GRS, the ORs are 3.4 for mild, 6.1 for moderate, and 9.2 for severe hearing impairment ($P = 3.0 \times 10^{-19}$, $8.4 \times 10^{-13}$ and $1.0 \times 10^{-7}$, respectively). Therefore, relative to the bottom GRS decile, the ARHI OR for carriers of rare variants is larger than the ARHI OR for individuals in the top GRS decile, but the ORs do not show significant heterogeneity ($P_{het} = 0.075$; Q-test). However, the risk of moderate and severe ARHI for carriers of rare variants is substantially greater than the risk for the top GRS decile ($P_{het} < 0.05$; Q-test).

As we have described, the severity of the hearing impairment for carriers of the highly penetrant variants in *LOXHD1* and *FBF1* varies from mild to profound. We hypothesize that the GRS could act as a modifier on the expressivity, i.e., that some of the variable expressivity of these highly penetrant variants could be explained by the common variants associating with ARHI. We estimated the relationship between these variants and the GRS on the PTA hearing thresholds and found positive interaction for both *LOXHD1* and *FBF1* ($P = 6.8 \times 10^{-4}$ and $P = 4.7 \times 10^{-3}$, respectively; likelihood-ratio test). This shows that among carriers of these highly penetrant genotypes, those who additionally have a high GRS are at a greater risk of a more severe ARHI than those that have a low GRS.

Long-term exposure to occupational loud noises is a risk factor for ARHI[9,45]. We had information on the occupation the subjects had for the majority of their lives for 7642 of the 11,484 individual in the DHS dataset. Three occupational categories associated with increased risk of ARHI; plant and machine operators and assemblers ($N = 508$, OR = 1.88, CI = [1.49;2.37], $P = 8.4 \times 10^{-8}$), craft and related trades workers ($N = 1,172$, OR = 1.56, CI = [1.32;1.84], $P = 1.3 \times 10^{-7}$) and agricultural and fishery workers ($N = 783$, OR = 1.55, CI = [1.30;1.85], $P = 1.6 \times 10^{-6}$). We tested for an interaction effect between long-term occupational noise exposure and the ARHI GRS on the risk of ARHI but did not find a significant interaction ($P = 0.94$; likelihood-ratio test). Among the individuals in the top GRS decile, noise exposure associates with increased risk of ARHI (OR = 1.77, CI = [1.18;2.67], $P = 6.3 \times 10^{-3}$) similar to the rest of the population (OR = 1.70, CI = [1.47;1.97], $P = 1.2 \times 10^{-12}$, $P_{het} = 0.86$; Q-test).

**Discussion**
In what we believe is the largest GWAS meta-analysis on ARHI to date, we found an association with 51 variants, of which 21 have not been reported before, using audiometric measurements from Icelanders and data on self-reported hearing difficulty from the UKB. This study yielded a larger number of rare variants, both under additive and recessive models, than previous GWAS studies that reported common variant associations with small to moderate effects on ARHI. These findings include variants in both known Mendelian deafness genes and genes not previously linked to hearing.

We constructed an ARHI GRS and found that individuals in the top GRS decile are at 2.5-fold greater risk than those in the bottom decile, and on average, they develop ARHI 10 years earlier than the bottom decile. The 2.5-fold greater risk is comparable to the 3.4-fold greater risk of carriers of rare ARHI variants than those in the bottom GRS decile. However, carriers of rare ARHI

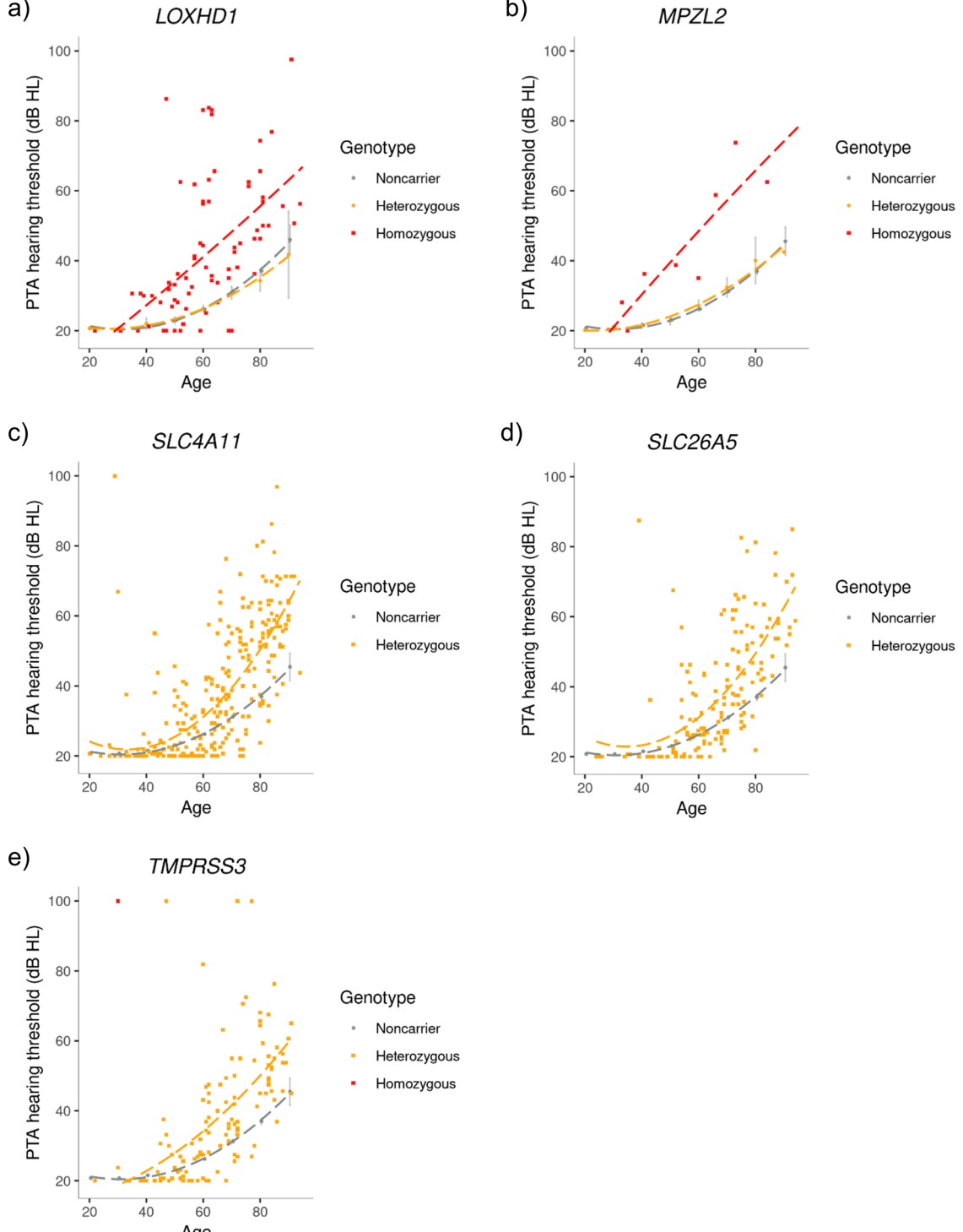

**Fig. 4 Changes in PTA hearing thresholds by age for carriers of rare ARHI variants in Mendelian deafness genes.** Effects of variants in **a** *LOXHD1*, **b** *MPZL2*, **c** *SLC4A11*, **d** *SLC26A5*, and **e** *TMPRSS3* are shown. In **a** and **b**, the average PTA in the DHS dataset are represented with gray dots for non-carriers and orange dots for heterozygotes and the PTA hearing thresholds of the homozygous carriers in DHS and NIHSI datasets are represented with red squares. In **c**, **d**, and **e**, the average PTA of non-carriers in the DHS dataset are represented with gray dots and the PTA hearing threshold of the heterozygous carriers in DHS and NIHSI datasets by yellow squares. The dashed red lines are linear regression lines of best fit and the dashed yellow and grey lines are polynomial regression lines of best fit. The gray vertical lines represent 95% confidence intervals. A figure for *TBC1D24* is not included because the variant was detected in UKB dataset only and therefore audiometric measures are not available for the carriers.

variants have substantially greater risk of moderate and severe ARHI than individuals in the top GRS decile, showing that the rare variants identified in this study predispose to more severe ARHI than the combination of common variants in the GRS.

Despite the importance of hearing in everyday life, ARHI is often not recognized by patients and left untreated; only 22% of people with mild hearing impairment report a hearing handicap[6]. Because of this, ARHI is often not diagnosed until

a)

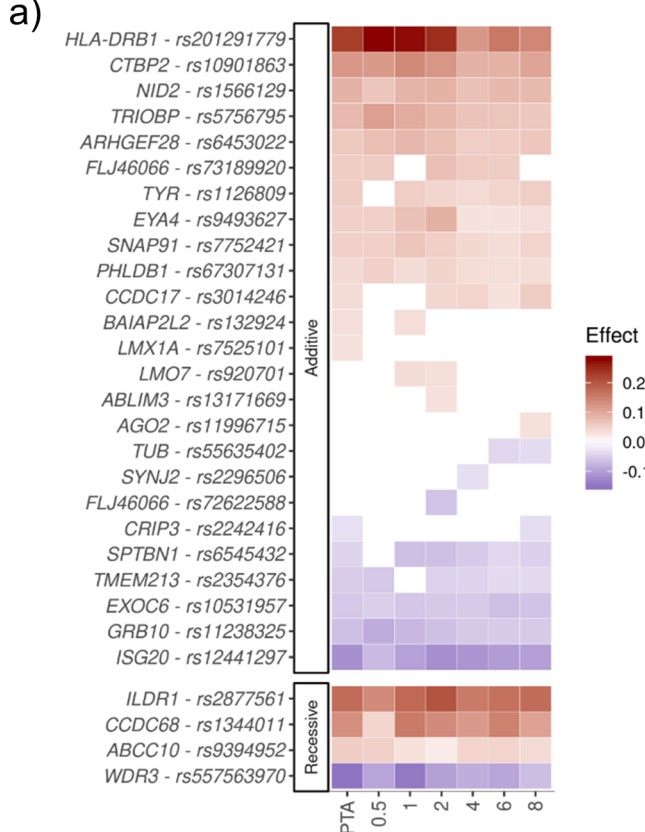

b)

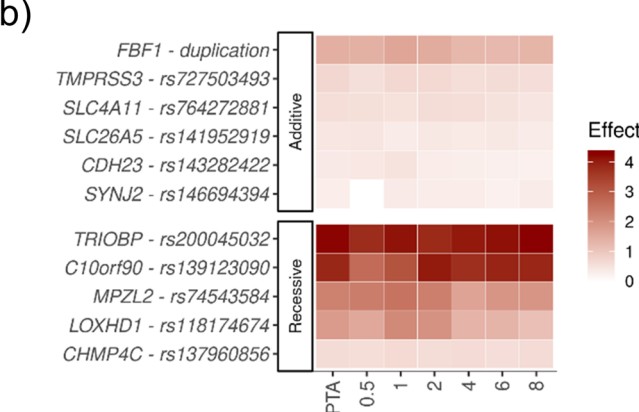

**Fig. 5 Effects of the ARHI variants on ARHI per frequency.** Each row shows the effect, the logarithm of the estimated odds ratios, of the minor allele on ARHI for PTA, the average of 0.5, 1, 2, and 4 kHz, and separately for each frequency, 0.5, 1, 2, 4, 6, and 8 kHz, for **a** common variants with EGF > 1% and **b** rare variants with EGF < 1%. The effect is shown only for associations with P-value <0.05. Red color represents an increased risk of ARHI and blue color represents decreased risk.

several years after onset and has then often already had many negative consequences such as effects on employment, social isolation, and depression[46]. Due to this, there is a need for better screening strategies, and using a GRS to stratify individuals into risk groups could enable enhanced screening. Identifying high-risk individuals might also help with preventing or reducing the severity of the hearing impairment. We have shown that noise exposure increases the risk of ARHI among those who are already at high genetic risk. It shows that avoiding loud noises is even more important for those who have a genetic predisposition to ARHI.

Previous reports have claimed that over 70% of non-syndromic prelingual hearing loss is inherited in a recessive manner[47]. In this study, we found six variants that associate with ARHI under a recessive mode of inheritance. For instance, the variant in *LOXHD1* is genome-wide significant in the UKB data alone, but was not detected by Wells et al. with the same dataset under an additive model[17]. In addition, we show that three variants previously reported to associate with ARHI under an additive model are truly recessive and four variants detected under an additive model in this study have stronger effects on homozygous carriers than expected under an additive model. These results highlight the importance of applying a recessive model when searching for variants associating with ARHI, which has not been done in previous GWASs.

A limitation of this work is that in the three datasets, the ARHI phenotype is defined in different ways. Because we did not restrict the definition of ARHI in terms of age of onset or severity, we performed follow-up analysis for all rare variants to make sure that the reported variants really associate with ARHI and not prelingual or childhood-onset deafness. Our results for common variants were mainly driven by the UKB dataset but 38 out of 51 variants replicated in the combined Icelandic datasets. The lack of replication in the DHS dataset is most likely due to smaller sample size, while in the NIHSI dataset it might be due to differences in the phenotype ascertainment, where patients are referred to NIHSI for hearing problems. Using population controls in the NIHSI dataset that have not been specifically screened for hearing impairment, will also misclassify some cases as controls. However, the effect sizes from the UK are highly correlated with effect sizes from Iceland and have consistent direction of effects. The age of onset, severity, and progression of ARHI are highly variable between individuals, and future GWAS could further analyze subtypes of ARHI. The Icelandic datasets provide more details regarding these factors as well as the measures of hearing at specific frequencies. Some ARHI variants have stronger effects on particular frequencies, while most affect all frequencies similarly.

We found six loci that have not been reported to affect hearing in humans before; *FBF1*, *FSCN2*, *TBX2*, *C10orf90*, *SH2D4B*, and *AP1M2*. Inner-ear protein expression analysis in mice[40] have shown that the mouse homologs Fscn2, Tbx2, C10orf90, Sh2d4b, and Ap1m2 have higher expression, ranging from 5 to 43-fold, in hair cells versus non-hair cells (Supplementary Table 6), suggesting that these genes have specialized roles in the inner-ear hair cells, but degeneration of the inner-ear hair cells is the main cause of ARHI[5].

The two strongest associations not reported before, were with the highly penetrant tandem duplication covering exons 4–7 in *FBF1*, detected under an additive model, and a missense variant in *LOXHD1*, affecting ARHI in homozygous state. *FBF1* encodes Fas-binding factor 1, a keratin-binding protein necessary for ciliogenesis[48,49]. We speculate that *FBF1* may have a role in the cilia of the inner ear, but further studies are needed to determine the biological effect of the duplication and the mechanism behind the association of *FBF1* with ARHI. *LOXHD1* encodes lipoxygenase homology domain 1, which consists of 15 PLAT (polycystin-1, lipoxygenase, alpha-toxin) domains[50]. Grillet et al. showed that *LOXHD1* is expressed in the functionally mature mechanosensory hair cells in the inner-ear and loss-of-function mutations in the gene lead to auditory defects in mice and humans, indicating an essential role for normal hair cell function[50]. Homozygous carriers of p.Arg1090Gln in *LOXHD1* report a younger age for hearing aid usage than the rest of hearing aid users, showing that the variant is associated with a severe form of ARHI. Given the frequency of p.Arg1090Gln (gnomAD, https://gnomad.broadinstitute.org/), we estimate the number of

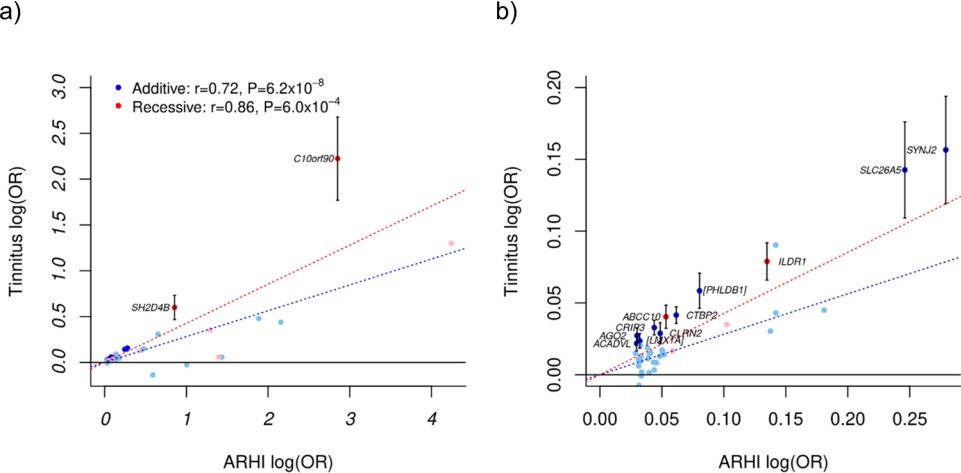

**Fig. 6 Effect of the ARHI variants on tinnitus.** The effect of the ARHI variants on ARHI is plotted against their effect on tinnitus for **a** all ARHI variants and **b** zoomed-in on variants from (**a**) with ARHI OR < 1.35. Effects are shown as the logarithm of the estimated odds ratios. ARHI variants detected under the additive model were tested for tinnitus using the additive model (blue dots) and ARHI variants detected under the recessive model were tested for tinnitus using the recessive model (red dots). The corresponding dashed lines represent results from a linear regression using MAF(1-MAF) as weights. Variants that affect tinnitus, controlling the false discovery rate at 0.05, are plotted with darker color and labeled with their corresponding gene. All effects are shown for the ARHI risk-increasing allele. Error bars represent 95% confidence intervals. The dotted lines represent results from a weighted linear regression using MAF(1-MAF) as weights, red for recessive variants and blue for additive, and the weighted correlation coefficients (*r*) and the corresponding *P*-values are shown in (**a**).

homozygous carriers to be around 300 in Iceland, 24,000 in the UK, and 300,000 in the whole of Europe.

We also tested the ARHI variants for association with tinnitus. Tinnitus is considered to have a broad etiology and can be caused by problems in the entire auditory pathway[51]. ARHI and tinnitus are correlated phenotypes, but shared genetic causes have not been broadly explored. We found that 13 of the 51 ARHI variants also associated with tinnitus, showing that some pathogenic processes that cause ARHI also increase the risk of tinnitus.

Our knowledge of the pathogenesis of ARHI is still limited, but the work presented here reveals several loci, shedding a light on the genetics underlying this common sensory defect.

## Methods

**Phenotype datasets.** For the meta-analysis, we conducted a GWAS of ARHI in three datasets under both additive and recessive models.

*The DHS dataset.* Pure tone audiometric air conduction testing was performed for 11,484 Icelanders as a part of a comprehensive phenotyping of a general population sample enriched for carriers of rare and potentially high impact mutations[25] (the deCODE health study[24]). Homozygous carriers of p.Arg1090Gln in *LOXHD1* were recruited, resulting in 37 carriers that participated (Supplementary Table 3). For the GWAS, 4140 individuals with PTA > 25 were defined as ARHI cases and the remaining 7344 as controls. Participation in the deCODE health study includes blood sample collection, numerous physical measurements, permission to access a wide range of health-related information including hospital data, a verbal interview, and an online questionnaire about health and lifestyle, including questions on hearing aid usage and tinnitus. All participants of the study gave written informed consent, in accordance with the Declaration of Helsinki, and the study was approved by the Icelandic Data Protection Authority and the National Bioethics Committee (VSNb2015120006/03.01 with amendments).

*The NIHSI dataset.* Pure tone audiometric air conduction testing was performed for 22,212 Icelanders at the National Institute of Hearing and Speech in Iceland (NIHSI). For the GWAS, 9619 individuals were defined as ARHI cases (PTA > 25) and 298,609 individuals were selected as population controls (excluding individuals in the DHS dataset). All participants who donated samples gave informed consent and the study was approved by the Icelandic Data Protection Authority and National Bioethics Committee (VSN-18-186).

*The UKB dataset.* The UKB study is a large prospective cohort study of around 500,000 individuals from the UK[52]. Extensive phenotypic and genotypic information has been collected for the participants, including self-reported hearing difficulty and tinnitus. For the GWAS, we defined 108,175 ARHI cases as those

who answered "Yes" or "I am completely deaf" to the question "Do you have any difficulty with your hearing?" and 285,746 controls as those who answered "No". In our analysis, we only included individuals determined to be of white British ancestry[53] and we use LD score regression[54] to account for inflation in test statistics due to relatedness. All participants of the UK Biobank study gave informed consent and the study was approved by the North West Research Ethics Committee (REC Reference Number: 06/MRE08/65).

**Audiometric test.** In DHS and NIHSI datasets, the pure tone air conduction audiometric test was performed by specially trained staff. The audiometer delivers pure tones at 0.5, 1, 2, 4, 6, and 8 kHz at different intensity levels, usually starting at 20 dB HL and increased if necessary. For each individual and each ear, the lowest intensity of sound detection is defined as their hearing threshold at that frequency. The pure tone average (PTA) was defined as the average hearing threshold at 0.5, 1, 2, and 4 kHz (according to the classification of the WHO). We define ARHI cases as those with PTA > 25.

**Genotype datasets.** In the Icelandic GWASs, using the DHS and NIHSI datasets, we analyzed high-quality 34.0 million sequence variants identified through whole-genome sequencing of 49,708 Icelanders which have been described in detail[55,56]. In summary, we whole-genome sequenced the Icelanders using Illumina technology to a mean depth of at least 17.8× and median depth of 36.9×. The sequence variants were jointly called using Graphtyper[57], thereof 79.318 high-confidence structural variants described previously[58]. We genotyped 166,281 Icelanders using Illumina SNP chips and their genotypes were phased using long-range phasing[59]. Genotypes of the 34.0 million sequence variants were imputed into all chip-typed Icelanders as well as relatives of the chip-typed, to increase the sample size for association analysis. All tested variants had imputation information over 0.8.

The UKB GWAS was performed with two sets of genotypes. The primary analysis was performed with 26.5 million high-quality variants (imputation info > 0.8) from the Haplotype Reference Consortium (HRC) reference panel, imputed into chip-typed individuals of European ancestry[53]. The genotyping was performed using a custom-made Affimetrix chip, UK BiLEVE Axiom in the first 50,000 individuals[60], and with Affimetrix UK Biobank Axiom array in the remaining participants[61]. Imputation was carried out by Wellcome Trust Centre for Human Genetics using a combination of 1000Genomes phase 3[62], UK10K[63], and HRC reference panels[64], for up to 93 million variants[53]. In addition, we performed a GWAS with 922 thousand variants identified through whole-exome sequencing of 49,960 study participants[65], imputed into chip-typed individuals of European ancestry.

**Statistics and reproducibility.** Logistic regression was used to test for the association between sequence variants and binary traits. For the additive model, the expected allele counts were used as a covariate while for the recessive model, the product of the maternal and paternal genotype probabilities was used as a covariate. For the genotypic model, separate parameters were included for heterozygotes and homozygotes. Other available individual characteristics that correlate with the trait were additionally

included in the model. In the DHS and NIHSI datasets, those were sex, county of birth, current age or age at death (including first and second-order terms), blood sample availability, and an indicator function for the overlap of the lifetime of the individual with the time span of phenotype collection. In the UKB dataset, those were sex, age, and 40 principal components in order to adjust for population stratification.

We used LD score regression to account for distribution inflation in the dataset due to cryptic relatedness and population stratification[54]. Using 1.1 million variants, we regressed the $\chi^2$ statistics from our GWASs against LD score and used the intercepts as a correction factor. The estimated correction factors for ARHI were 1.05, 1.20, and 1.05 in DHS, NIHSI, and UKB datasets, respectively.

Because the UKB GWAS was performed on two sets of genotypes, we performed two separate meta-analyses. Both meta-analyses combined results from three GWAS using DHS, NIHSI, and UKB datasets. In meta-analysis I, we used the UKB GWAS results based on the variants from the HRC reference panel and in meta-analysis II we used the UKB GWAS results based on the variants identified through whole-exome sequencing. When meta-analyzing the three GWASs, we used a fixed-effects inverse variance method[66] which is based on effect estimates and standard errors from all datasets. Sequence variants from Iceland and the UKB were matched on position and alleles.

For the genotypic model $P$-values were computed by comparing the genotypic model to the null model, using a likelihood-ratio test. For the genotypic model meta-analysis, sample size approach was used based on $P$-values and sample size[67].

A Q-test[68] was used to test for heterogeneity between effect sizes.

**Definition of ARHI variants**. The PTA-based definition of ARHI used in the Icelandic datasets does not exclude individuals that are completely deaf or have childhood-onset hearing loss. The GWAS can therefore detect rare associating variants that cause prelingual or childhood-onset hearing loss instead of ARHI. Due to this, for all the rare variants that satisfied the genome-wide significance thresholds, we fit a linear regression model, with the PTA hearing threshold of the carriers as response and age as covariate, to estimate the predicted PTA hearing threshold of the carriers in childhood. Variants that had predicted hearing threshold of 25 dB HL at 10 years of age were considered to be causing childhood-onset hearing loss.

**Significance thresholds**. The genome-wide significance thresholds were corrected for multiple testing with a weighted Bonferroni adjustment[32]. The weights, based on enrichment of variant classes with predicted functional impact among association signals, were estimated from the Icelandic data, resulting in significance thresholds of $2.4 \times 10^{-7}$ for loss-of-function variants, $4.9 \times 10^{-8}$ for moderate-impact variants, $4.4 \times 10^{-9}$ for low-impact variants, $2.2 \times 10^{-9}$ for other variants within DHS sites and $7.4 \times 10^{-10}$ for remaining variants.

We evaluated false discovery rate, assessed with the $q$-value package in R. The $P$-value cutoff of $5.0 \times 10^{-8}$ corresponded to $q$-values of 0.0013 for the additive model and 0.0025 for the recessive model, which add up to 0.4%.

In the burden test, a genome-wide significance threshold of $0.05/18,482 = 2.7 \times 10^{-6}$ was used, correcting for the number of autosomal protein-coding RefSeq genes[69,70].

**Conditional analysis**. To search for secondary association signals at each locus, we applied a stepwise conditional analysis, adding the top variant as a covariate when testing all other variants in a 1 Mb window around the top variant. We used a Bonferroni-adjusted significance threshold for secondary associations. We found independent secondary associations at 6 loci.

**Genetic risk score**. The GRS for ARHI was constructed using the 35 detected variants with EGF > 1% and estimated effects from the UKB dataset. If we let $m_{vi}$ and $p_{vi}$ be the genotype probability for individual $i$ and sequence variant $v$ at the maternally and paternally inherited chromosomes, the GRS for individual $i$ is defined as

$$grs_i = \sum_{v=1}^{n} (m_{vi} + p_{vi})\beta_v + \sum_{v=1}^{m} (m_{vi} \times p_{vi})\gamma_v,$$

where $\beta$ are the effects of the $n$ variants detected with the additive model and $\gamma$ are the effects of the $m$ variants detected with the recessive model.

**Correlation between effect sizes**. When assessing the relationship between effect sizes, we fitted a weighted linear regression model where each variant was weighted by $f(1-f)$ where $f$ is the minor allele frequency of the variants.

**Reporting summary**. Further information on research design is available in the Nature Research Reporting Summary linked to this article.

## Data availability
The authors declare that the data supporting the findings of this study are available within the article, in supplementary files, and upon request. The sequence variants from the Icelandic population whole-genome sequence data have been deposited at the European Variant Archive under accession PRJEB15197. GWAS summary statistics will be made available at http://www.decode.com/summarydata.

## Code availability
We used the following publicly available software for the whole-genome sequencing process: BWA 0.7.10 mem (https://github.com/lh3/bwa), Picard tools 1.117 (https://broadinstitute.github.io/picard/), SAMtools 1.3 (http://samtools.github.io/), Bedtools v2.25.0-76-g5e7c696z (https://github.com/arq5x/bedtools2/), GraphTyper 1.3 (https://github.com/DecodeGenetics/graphtyper), Variant Effect Predictor (https://github.com/Ensembl/ensembl-vep).

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

## Acknowledgements

We would like to thank all study participants for their valuable contribution to the research. We also thank the staff at NIHSI and deCODE genetics recruitment center and core facilities, and all our colleagues, for their important collaboration on this work. A part of this research has been conducted using the UK Biobank Recourse under application number 56270.

## Author contributions

E.V.I., H.H., S.B., U.T., P.S., I.H., I.J., D.F.G., and K.S. designed the study and interpreted the results. E.V.I. S.B., G.S., G.Thorleifsson, H.P.E., G.H.H., K.E.H., P.M., A.G., A.O., G.A.A., B. O.J., L.S., B.H., and D.F.G. analyzed the data. As.J. and Ad.J. did the Sanger sequencing. E.V. I., H.H., G.Thorleifsson, T.J., V.T., H.P., G.Thorgeirsson, I.H., and I.J. performed recruitment and phenotyping. The manuscript was drafted by E.V.I., H.H., S.B., T.O., U.T., P.S., I. H., D.F.G., and K.S. All authors contributed to the final version of the manuscript.

## Competing interests

E.V.I., H.H., S.B., T.O., G.S., G.Thorleifsson, H.P.E., G.H.H., K.E.H., P.M., A.G., G.A.A., A.O., B.O.J., As.J., Ad.J., T.J., L.S., V.T., B.V.H., G.Thorgeirsson., U.T., P.S., I.J., D.F.G., and K.S. are employees of deCODE genetics/Amgen, Inc. H.P. and I.H. have no competing interests to declare.
