## [Transparent Peer Review File · Communications Biology]

Reviewers' comments:

Reviewer #1 (Remarks to the Author):

Key results

The authors present a very nice study on ARHI, which is a great addition to the existing literature. It is generally well written, concise where needed, and the clarity of the text is good. It is clear that the authors are experts in the field of genetics. Data interpretation seems valid and justify their concluding statements. I think this manuscript is definitely fit for this journal, but may need several (minor) revisions, listed below.

Please indicate any particular part of the manuscript, data, or analyses that you feel is outside the scope of your expertise, or that you were unable to assess fully.

I am an ENT-surgeon with interest in the genetics of ARHI. My expertise lies in hearing, the cochlea, and genes involved. I am not a geneticist by training, so I cannot fully comment on the validity of the methods of genetics nor its computational analyses.

References

Add: Nagtegaal et al. Genome-wide association meta-analysis identifies five novel loci for age-related hearing impairment. Sci Rep. 2019 Oct 23;9(1):15192. doi: 10.1038/s41598-019-51630-x. To page 3: Genome wide association studies (GWAS) on ARHI have been performed lines 10–15

Revisions:

Page 2:

- ARHI is one of the most common chronic conditions (s missing in text)

Page 3, 2nd paragraph:

- Normal hearing threshold: the definition of "normal" is dependent on age and frequency.
- PTA: the included frequencies depend on definition. Maybe rephrase: "Hearing thresholds at frequencies 0.5, 1, 2 and 4 kHz were used in a pure tone average, according to the WHO classification of hearing loss. These frequencies represent the range of speech."

Page 4:

- DHS dataset:

o The subjects were between 18 and 97 years of age at time of recruitment. When you're studying ARHI, wouldn't it be good to set a minimum age? People aged 20 – 40 without hearing loss cannot be seen as controls, as hearing loss might not have occurred yet
- Due to this bias, we defined the 9,619 subjects with PTA above 25 dB HL as ARHI cases and designated 298,609 Icelanders with no available hearing data as population controls (excluding individuals in the DHS dataset).

o Please clarify: you've designated participants to controls where there were no hearing data available? There are bound to be many people with ARHI in that group as well?

Page 5:

- The section on ARHI and reduced height seems a bit out of place in this (genetic) paper. Can you relate the association between reduced height and ARHI with any known genes associated with short stature?

Page 15:

- BFB1 is not related to hearing loss in mice:

https://www.mousephenotype.org/data/charts?accession=MGI:1922033&allele_accession_id=MGI:4432655&meter_stable_id=IMPC_ABR_002_001&zygosity=homozygote&phenotyping_center=WTSI&pipeline_stable_id=MGP_001

Do the authors have an explanation for this discrepancy?

Page 16:

- For the meta-analysis we conducted a GWAS of ARHI in three datasets under both an additive and a recessive models. Remove "s" in models (or "an" and "a").
- The NIHSI dataset: already mentioned, how were the controls selected? Were they randomly taken from the population? How do you know they do not have hearing loss?

Page 17:

- On PTA: in general, you can include any frequency in the PTA. So I would advise to rephrase slightly: The pure tone average (PTA) was defined as the average hearing threshold at 0.5, 1, 2 and 4 kHz (according to the classification of the WHO).

Page 19:

- "child-hood", replace by childhood

Page 20:

- A statement on (summary) data availability or a link to a repository should be included, as is common in genetics nowadays.

Suppl notes

Page 10:

- "We performed Sanger sequenced.." Replace by: We performed Sanger sequencing

Suppl table 10:

- The GFP+ cells include hair cells (sensory epithelium) from the cochlea but also the utricle, which is part of the vestibular system. So this measure does not fully indicate cochlear expression. For example, CRIP3 is mainly expressed in the utricle and less so in the cochlea. So the GFP+/GFP- ratio is not a perfect measure for specific cochlear expression. Second, genes affecting supporting cells (GFP-) can also cause hearing loss.
- Fold change is reversed. It should be listed as GFP+/GFP-
- SLC26A5: FDR seems to be listed incorrect. The SHIELD website gives a FDR of 9.96e-10.

Suppl table 11:

- I'm not sure I fully understand this table. The OR listed in the last column refers to the odds of having tinnitus when a person has that specific gene variant, is that correct? Most of the OR's are around 1. Some of the significant variants have OR's of 1.02 – 1.04. That means there's only a very small effect on having tinnitus. How would you interpret these OR's? Could you add a bit of text to the legend?

The authors present a very nice study on ARHI, which is a great addition to the existing literature. It is generally well written, concise were needed, and the clarity of the text is good. It is clear that the authors are experts in the field of genetics. Data interpretation seems valid and justify their concluding statements. I think this manuscript is definitely fit for this journal, but may need several (minor) revisions.

AP Nagtegaal

Reviewer #2 (Remarks to the Author):

Ivarsdottir et al present within their manuscript "The genetic architecture of age-related hearing impairment" a meta-analysis of hearing loss in 3 genetic cohorts in which they report 46 loci associated with age related hearing loss including 22 novel associations. Two of these cohorts are Icelandic cohorts of 4,140 cases and 9,629 cases both with pure tone audiometry and whole genome sequencing data. The third is the much larger UK Biobank cohort with 108,175 cases of self-reported

hearing loss on which the authors analyse both gene SNP data and whole exome sequencing data.

GWAS on the UK Biobank SNP data and self-reported hearing has been published previously but only under the additive model, here the authors also analyse the data under the dominant and recessive model and include rare variant analyses. There are very few publications on genetics of age-related hearing impairment that report significant findings so these findings would be very welcome. However, there are several major flaws in the design and data used within the study which raise serious questions about the findings. These are:

(1) There are known errors with the WES data from UKBB used here and these have been known since 2019 see <https://www.ukbiobank.ac.uk/wp-content/uploads/2020/06/UKBiobank-50k-Exome-Release-FAQ-June-2020.pdf>

These issues will impact any study using them in associations, especially in rare variant analyses where there are smaller numbers involved, potentially leading to spurious associations. Given how long this WES data has been available very few studies have been published due to these errors and I do not

consider them a reliable dataset. An updated WES on 200,000 individuals is expected to be released by the end of the year.

(2) The nature of the three different cohorts is very different which makes replication difficult and the definition of ARHI is problematic. One Icelandic cohort has an age range of 18-97 and then other contains 43% children. These are not ideal cohorts to study an age-related disease that predominantly affects over 50s and the authors have not used age as a factor in their definition of ARHI, other than to exclude under 10s. In the UK Biobank which is more pertinent to ageing research (40-69 years) they have not excluded individuals who describe themselves as "completely deaf" a subset that will include individuals who have a congenital deafness.

(3) Their analyses focuses on the recessive and dominant models rather than the additive that would normally be applied to a common, complex disease like ARHI. This would seem to bias the analyses towards finding monogenic effects and when combined with (2) is likely to favour detection of congenital deafness genes in younger people rather than ARHI genes. This makes the interpretation of the study problematic.

(4) A further issue is the multiple testing corrections which have been applied. The definition of genome-wide significance is non-typical and it is unclear whether the authors have taken into consideration they have tested more than one genetic model in their Bonferroni correction?

Reviewer #3 (Remarks to the Author):

Age related hearing impairment (ARHI) and tinnitus are serious and disabling conditions which are both heritable. This manuscript describes work using large datasets to perform the largest study to date, which advances our understanding of the genetic mechanisms underlying both conditions.

The study has used two samples from deCODE, Iceland which have been phenotyped using pure tone audiometry in air (PTA), the gold standard method of detecting hearing loss, and whole genome sequenced. The DHS subsample was enriched for carriers of rare mutations (4k cases with pure tone average >25dB and 7k controls) while the NIHSI subsample is highly skewed towards ARHI so cases (10k PTA >25dB) were matched with non-phenotyped population controls (almost 300k). The UKB samples, on the other hand, used self-reported hearing loss in a population sample with inadequate hearing tests and a mixture of imputed GWAS and whole exome sequencing. Two meta-analyses were performed using the same overall case control mix but different genotyping approaches.

The results reveal a number of variants which have already been identified using the UKB dataset but importantly find 22 novel variants, 14 of which are rare and 6 are novel associations with hearing loss. In addition, the authors found a rare missense variant – a tandem duplication – having MAF 2% which confers an odds ratio of 4.2, which they included in a genetic risk score. The paper discusses the differences between the Icelandic and UK findings including the different genetic structures and is very well written. The manuscript is clear in its methods and the presentation of the results and appropriately adjusted for multiple testing. The findings of additional variants through application of the recessive model is of interest. There is a great deal of work here and a lot of results of importance to the hearing field in general. Consideration should be given to submitting the tinnitus/ARHI in a separate manuscript.

Major comments

1. The biggest challenge in a study such as this is to convince the reader that a study combining PTA in a genetically isolated population with a self-reported hearing loss measure in a heterogeneous population produces variants that are associated with ARHI. Here the definition of ARHI is problematic, and the study assumes that hearing loss is age-related when we know that the Icelandic samples are mixed in origin as far as deafness goes. So while the associations reported may be real, they beg the question of what phenotype the association is actually with. For those variants in both countries' datasets this is relatively straightforward but this challenge touches on the issue of the heterogeneity of ARHI. There is a no limitation paragraph to the discussion – please add one.
2. Leading on from this, the Discussion should include consideration of potential for subtyping ARHI in future.
3. While Figure 1 is a clear flow diagram of the approach taken, the meta-analyses I and II are not described clearly in the Methods section. Please provide details
4. The association with tinnitus – also a highly important and disabling phenotype - appears to be a bolt-on. While the subject is covered in the Introduction it doesn't get a mention in the abstract and the data are hidden in Supplementary Table 11. Shared genetic variants between the two phenotypes are identified (n=13) that ARHI risk alleles increase risk of tinnitus. Please give suitable prominence throughout all sections.
5. Noise exposure is an important consideration and the DHS dataset is very useful in showing the influence of occupation to the risk of ARHI. However if the top GRS decile shows no increase risk with noise (P13 line 1) how do the authors justify suggesting the use of GRS as a screening tool and recommend noise avoidance (p14 line 1)?
6. There is further inconsistency around noise exposure in Discussion paragraph 3. If only 22% of people with mild ARHI report hearing handicap then again genetic screening makes no sense – hearing screening is what is indicated to demonstrate to an individual that there is actual hearing loss present not an abstract risk of it. Please amend

Minor comments:-

1. The abstract is a poor reflection of the totality of the study, with methods lacking. Please include the phenotyping approach used in the various cohorts.
2. Please make clear in Methods the difference between meta-analysis I and II
3. Please add more detail around the patients included in the NIHSI dataset – are these people referred with hearing and speech difficulties?
4. Please add the tinnitus methods and results to abstract
5. Typo – page 7 line 3 MAF<10.0%

Reviewer #1 (Remarks to the Author):

Key results

The authors present a very nice study on ARHI, which is a great addition to the existing literature. It is generally well written, concise were needed, and the clarity of the text is good. It is clear that the authors are experts in the field of genetics. Data interpretation seems valid and justify their concluding statements. I think this manuscript is definitely fit for this journal, but may need several (minor) revisions, listed below.

Please indicate any particular part of the manuscript, data, or analyses that you feel is outside the scope of your expertise, or that you were unable to assess fully.

I am an ENT-surgeon with interest in the genetics of ARHI. My expertise lies in hearing, the cochlea, and genes involved. I am not a geneticist by training, so I cannot fully comment on the validity of the methods of genetics nor its computational analyses.

References

Add: Nagtegaal et al. Genome-wide association meta-analysis identifies five novel loci for age-related hearing impairment. Sci Rep. 2019 Oct 23;9(1):15192. doi: 10.1038/s41598-019-51630-x. To page 3: Genome wide association studies (GWAS) on ARHI have been performed lines 10–15

Response: We have now added this paper as a reference.

Revisions:

Page 2:

- ARHI is one of the most common chronic conditions (s missing in text)

Response: This has now been fixed.

Page 3, 2nd paragraph:

- Normal hearing threshold: the definition of "normal" is dependent on age and frequency.

Response: We removed this sentence from the manuscript without losing valuable information.

- PTA: the included frequencies depend on definition. Maybe rephrase: "Hearing thresholds at frequencies 0.5, 1, 2 and 4 kHz were used in a pure tone average, according to the WHO classification of hearing loss. These frequencies represent the range of speech."

Response: We have now rephrased this sentence as the reviewer suggests:

p.3 "The lowest intensity of sound detection for each individual is defined as their hearing threshold. According to the WHO classification of hearing loss, subjects with a hearing threshold above 25 dB HL are considered to have hearing impairment and the higher the thresholds the greater the impairment¹⁸ (Supplementary Table 1). Hearing thresholds at frequencies 0.5, 1, 2 and 4 kHz were used in a pure tone average (PTA). These frequencies represent the range of speech."

Page 4:

- DHS dataset:

o The subjects were between 18 and 97 years of age at time of recruitment. When you're studying

ARHI, wouldn't it be good to set a minimum age? People aged 20 – 40 without hearing loss cannot be seen as controls, as hearing loss might not have occurred yet

Response:

We note that 84% of the subjects in the DHS dataset are older than 40 years old and to account for age in the association testing, we adjust for age and age². To provide more detail of the age distribution, we have now added the figures below as Supplementary Figure 1, which show the YOB for cases and controls.

Due to the concern of the reviewer, we investigated the impact of including young controls in the GWAS, by performing the GWAS for the DHS and NIHSI datasets again after removing individuals that are younger than 40 years old at time of measurement. We have now added these results to Supplementary Table 12. Here below are figures that show the effect sizes with and without subjects less than 40 years old. The black solid lines are 45-degree lines, while red lines show result from a linear regression between the variables on the x and y axis.

Effect sizes for common variants:

log(OR), NIHSI dataset over 40 yo

log(OR), DHS dataset over 40 yo

P-values for common variants:

-log10 P, NIHSI dataset over 40 yo

-log10 P, DHS dataset over 40 yo

Effect sizes for rare variants:

log(OR), NIHSI dataset over 40 yo

log(OR), DHS dataset over 40 yo

P-values for rare variants:

-log10 P, NIHSI dataset over 40 yo

-log10 P, DHS dataset over 40 yo

These analyses show that removing younger individuals has essentially no impact on the effect estimates for the common variants. However, for the rare variants in *TRIOBP* and *C10orf90*, we remove a substantial part of the carriers and therefore lose power to detect the associations.

After this investigation we conclude that using all available information is the better option since it increases power to detect rare variants that cause a more severe form of ARHI, and does not bias results for common variants.

- Due to this bias, we defined the 9,619 subjects with PTA above 25 dB HL as ARHI cases and designated 298,609 Icelanders with no available hearing data as population controls (excluding individuals in the DHS dataset).

o Please clarify: you've designated participants to controls where there were no hearing data available? There are bound to be many people with ARHI in that group as well?

Response (*): The controls are selected from individuals in the deCODE genotype database (both the set of chip-typed individuals and family imputed individuals using the genealogy database), matched for age, that had never been referred to the NIHSI and did not participate in the DHS.

Using available genotyped individuals as controls, even though they have not been specifically screened for the trait of interest, is common in GWASs in order to increase statistical power (Mitchell et al.). We are aware that by using this approach, we will misclassify some cases as controls, and we have now added this point in the limitation section in the Discussion:

p.15 "The lack of replication in the DHS dataset is most likely due to smaller sample size, while in the NIHSI dataset it might be due to differences in the phenotype ascertainment, where patients are referred to NIHSI for hearing problems. Using population controls in the NIHSI dataset that have not been specifically screened for hearing impairment, will also misclassify some cases as controls. However, the effect sizes from the UK are highly correlated with effect sizes from Iceland and have consistent direction of effects."

Page 5:

- The section on ARHI and reduced height seems a bit out of place in this (genetic) paper. Can you relate the association between reduced height and ARHI with any known genes associated with short stature?

Response: The combination of hearing loss and short stature is common in many genetic syndromes (Barrenäs et al.). We investigated whether there is a correlation between the effect of the ARHI variants on ARHI and their effect on height and we did not find a significant correlation ($r=-0.18$, $P=0.13$). We also evaluated the correlation between the effect on height in the Icelandic and UK Biobank data ($N = 490,381$) of 693 reported adult height variants (Wood et al.) and their effects on ARHI and found no correlation ($r=0.014$, $P=0.72$).

Page 15:

- FBF1 is not related to hearing loss in mice:

https://www.mousephenotype.org/data/charts?accession=MGI:1922033&allele_accession_id=MGI:4432655¶meter_stable_id=IMPC_ABR_002_001&zygosity=homozygote&phenot

yping_center=WTSI&pipeline_stable_id=MGP_001
Do the authors have an explanation for this discrepancy?

Response: Although mouse models are a valuable tool for studying genes that affect hearing in humans (Angeli et al.), some genes that are known to play an essential role in hearing in humans have not been shown to affect hearing in mice (Makishima et al.; Parker et al.). More generally, human phenotypes are only recapitulated in mice around 40% of the time (Meehan et al.; Cacheiro et al.).

As we mention in the discussion, further studies are needed to determine the biological effect of the duplication in *FBF1* and to understand the mechanism behind the association of *FBF1* with ARHI.

Page 16:

- For the meta-analysis we conducted a GWAS of ARHI in three datasets under both an additive and a recessive models. Remove "s" in models (or "an" and "a").

Response: This has now been fixed.

- The NIHSI dataset: already mentioned, how were the controls selected? Were they randomly taken from the population? How do you know they do not have hearing loss?

Response: See the response to the similar comment above (*).

Page 17:

- On PTA: in general, you can include any frequency in the PTA. So I would advise to rephrase slightly: The pure tone average (PTA) was defined as the average hearing threshold at 0.5, 1, 2 and 4 kHz (according to the classification of the WHO).

Response: We have now rephrased this sentence as the reviewer suggested.

Page 19:

- "child-hood", replace by childhood

Response: This has now been fixed.

Page 20:

- A statement on (summary) data availability or a link to a repository should be included, as is common in genetics nowadays.

Response: We have now added a data availability statement to the manuscript.

Suppl notes

Page 10:

- "We performed Sanger sequenced.." Replace by: We performed Sanger sequencing

Response: This has now been fixed.

Suppl table 10:

- The GFP+ cells include hair cells (sensory epithelium) from the cochlea but also the utricle, which is part of the vestibular system. So this measure does not fully indicate cochlear expression. For example, CRIP3 is mainly expressed in the utricle and less so in the cochlea. So the GFP+/GFP- ratio is not a perfect measure for specific cochlear expression. Second, genes affecting supporting cells (GFP-) can also cause hearing loss.

Response: In the SHIELD paper, they found that hearing loss genes are highly enriched for differential expression in hair cells versus non-hair cells (Shen et al.). The following is a paragraph from that paper:

“Among the most differentially expressed genes, we found that homologs of established human hearing loss genes are highly enriched. Thirty-three of the 72 well-established hearing loss genes are differentially expressed by at least 2- fold with a false discovery rate (FDR) of <0.1, but only 6.8% of all genes meet the same criteria (odds ratio = 6.6, 95% confidence interval 4.3–10.0, Z = 8.9, P < 0.0001). This suggests that the likelihood of a gene’s impact on inner ear function can be estimated based on the degree (fold change) and statistical significance (FDR) of the differential expression in hair cells versus non-hair cells.”

We agree that this measure does not fully indicate a specific cochlear expression, but high expression in hair cells vs non-hair cells suggests that the gene has some specific role in the inner ear hair cells. We agree with the reviewer that this type of evidence is not conclusive, and have therefore tried to make clear that this is suggestive evidence for a mechanism. Additionally, we have now added the fold change for Cochlea/Utricle expression to Supplementary Table 10.

- Fold change is reversed. It should be listed as GFP+/GFP-

Response: This typing error has now been fixed.

- SLC26A5: FDR seems to be listed incorrect. The SHIELD website gives a FDR of 9.96e-10.

Response: This typing error has now been fixed.

Suppl table 11:

- I’m not sure I fully understand this table. The OR listed in the last column refers to the odds of having tinnitus when a person has that specific gene variant, is that correct? Most of the OR’s are around 1. Some of the significant variants have OR’s of 1.02 – 1.04. That means there’s only a very small effect on having tinnitus. How would you interpret these OR’s? Could you add a bit of text to the legend?

Response: We performed a GWAS for self-reported tinnitus from the DHS and UKB. The OR are the estimated OR from meta-analyzing these two datasets. We have now added this information to the legend to clarify this. We interpret these ORs as we normally interpret ORs from GWAS, where it is normal for common variants to have small effects.

Reviewer #2 (Remarks to the Author):

Ivarsdottir et al present within their manuscript “The genetic architecture of age-related hearing impairment” a meta-analysis of hearing loss in 3 genetic cohorts in which they report 46 loci associated with age related hearing loss including 22 novel associations. Two of these cohorts are Icelandic cohorts of 4,140 cases and 9,629 cases both with pure tone audiometry and whole genome sequencing data. The third is the much larger UK Biobank cohort with 108,175 cases of self-reported hearing loss on which the authors analyse both gene SNP data and whole exome sequencing data. GWAS on the UK Biobank SNP data and self-reported hearing has been published previously but only under the additive model, here the authors also analyse the data under the dominant and recessive model and include rare variant analyses. There are very few publications on genetics of age-related hearing impairment that report significant findings so these findings would be very welcome. However, there are several major flaws in the design and data used within the study which raise serious questions about the findings. These are:

(1) There are known errors with the WES data from UKBB used here and these have been known since 2019 see <https://www.ukbiobank.ac.uk/wp-content/uploads/2020/06/UKBiobank-50k-Exome-Release-FAQ-June-2020.pdf>

These issues will impact any study using them in associations, especially in rare variant analyses where there are smaller numbers involved, potentially leading to spurious associations. Given how long this WES data has been available very few studies have been published due to these errors and I do not consider them a reliable dataset. An updated WES on 200,000 individuals is expected to be released by the end of the year.

Response: UK biobank recently released an update of the WES data with 200K individuals and we used the new data to investigate the 5 variants that we found to associate with ARHI. In the table below, we calculated how many of the sequenced carriers, according to the 50K WES data, were confirmed as carriers in the 200K WES data (true carriers). We also assessed the quality of the imputation by calculating how many of the imputed carriers are sequenced carriers in the 200K data (true imputed carriers) and how many of the imputed non-carriers are carriers in the 200K WES data (false imputed non-carriers).

Variant	MAF	true carriers	true imputed carriers	false imputed non-carriers
chr5:272748	0.09%	78%	75%	0.0051%
chr16:2497068	0.01%	100%	100%	0.0029%
chr22:50549067	0.06%	100%	100%	0.0023%
chr21:42389042	0.07%	99%	98%	0.030%
chr11:118262596	0.83%	100%	100%	0.00057%

We note that none of the 5 variants are ultra-rare, where the variants with the lowest MAF (chr16:2497068 in *TBC1D24*) has 12 sequenced carriers.

We also performed additional QC procedures to further validate the 5 variants. Leave-one-out r^2 is defined as r^2 between the observed genotypes from sequencing and genotypes imputed into each sequenced individual from all the other sequenced individuals. The leave-one-out r^2 was over 0.8 for the 5 variants:

Variant	MAF	leave-one-out r^2
chr5:272748	0.09%	0.93
chr16:2497068	0.01%	0.82
chr22:50549067	0.06%	0.98
chr21:42389042	0.07%	0.84
chr11:118262596	0.83%	0.95

We also looked at the allelic balance for the 5 variants, defined as the fraction of reads containing the alternative allele out of the reads containing the alternative and reference alleles combined. The following histograms show the allelic balance for the variants:

The variant in *PDCD6* has only 75% of its imputed carriers confirmed as carriers in the new WES data and its' allelic balance distribution is shifted and is not centered at 0.5. Due to this, we think it is possible that this variant is affected by the known duplicate read marking issue and we have therefore removed this variants from the manuscript.

However, these analyses show that there is no evidence that the other four variants are affected by the duplicate read marking issue and we are confident that the results we report for them are valid.

(2) The nature of the three different cohorts is very different which makes replication difficult and the definition of ARHI is problematic. One Icelandic cohort has an age range of 18-97 and then other contains 43% children. These are not ideal cohorts to study an age-related disease that predominantly effects over 50s and the authors have not used age as a factor in their definition of ARHI, other than to exclude under 10s. In the UK Biobank which is more pertinent to ageing research (40-69 years) they have not excluded individuals who describe themselves as "completely deaf" a subset that will include individuals who have a congenital deafness.

Response: See response to (3) below.

(3) Their analyses focuses on the recessive and dominant models rather than the additive that would normally be applied to a common, complex disease like ARHI. This would seem to bias the analyses towards finding monogenic effects and when combined with (2) is likely to favour detection of congenital deafness genes in younger people rather than ARHI genes. This makes the interpretation of the study problematic.

Response to (2) and (3):

We used the additive model and recessive model, not the dominant model. We state that we use the additive model multiple times throughout the paper and we never mention a dominant model.

Regarding the age distribution in the Icelandic cohorts, we have now added figures in the Supplement to show the year of birth distributions of the NISHI and DHS datasets (also shown above in response to comment from reviewer #1). Even though the NISHI dataset contains 43% children, we use very few children as cases in the GWAS, and 93% of defined cases had their hearing measured after the age of 40. We also performed the GWAS again using only individuals over 40 years old and found it to have no impact on effects and P-values for common variants, while lowering the power to detect some of the rare variants (see response to comment from reviewer #1).

We agree that it would be ideal to use age in the definition of ARHI, but in practice it would be difficult given the data we have, since age of onset is not documented in all datasets, and only a subset in the NISHI dataset have multiple measures for different time points. Therefore, we decided to define ARHI cases in the Icelandic datasets as those with measured PTA > 25 no matter the severity or age at time of measure.

Using this approach, we found many common variants associating with this definition of ARHI and we can assume that common variants are not causing congenital deafness. However, because we do not limit the definition of ARHI with respect to age or severity, rare variants detected might be variants causing congenital deafness. For the rare variants we detected, we estimated whether they were causing congenital or childhood-onset hearing impairment by analyzing the relationship between the

measured hearing thresholds and age. Four were indeed associated with childhood onset hearing loss and we therefore did not report them as ARHI variants (see p.6, Supplementary Table 7).

We have now emphasized this procedure in the Result section, p.6 as well as adding this to the Discussion:

p.6 “Because we do not restrict the definition of ARHI cases with respect to age at measure or severity, we might detect rare variants in the meta-analysis that are causing prelingual or childhood-onset hearing loss instead of ARHI. Due to this, we used the audiometric measures in the Icelandic datasets to estimate the predicted hearing threshold of the carriers in childhood and observed that 4 of the 56 variants cause prelingual or childhood-onset hearing loss rather than ARHI (Methods, Supplementary Table 7 and Supplementary Note 1).”

p.15 “A limitation of this work is that in the three datasets, the ARHI phenotype is defined in different ways. Because we did not restrict the definition of ARHI in terms of age of onset or severity, we performed follow-up analysis for all rare variants to make sure that the reported variants really associate with ARHI and not prelingual or childhood-onset deafness.”

(4) A further issue is the multiple testing corrections which have been applied. The definition of genome-wide significance is non-typical and it is unclear whether the authors have taken into consideration they have tested more than one genetic model in their Bonferroni correction?

Response: The typical genome-wide threshold of 5×10^{-8} assigns equal prior probability of association to all the sequence variants that are tested. We use a weighted Bonferroni correction method (Sveinbjornsson et al.) that corrects for the number of tested variants and uses enrichment of sequence annotations among association signals as weights. This method has been shown to increase power to detect associations. We note that, for high impact variant we use a slightly less stringent threshold than the traditional threshold, while for the majority of variants we use a more stringent threshold, since the threshold of 5×10^{-8} does not properly account for the number of sequence variants tested.

In the discovery study, we think it would be overly conservative to correct for the number of models used. The number of significant findings for each model is substantial which translates into a false discovery rate of 0.0013 for the additive model and 0.0025 for the recessive model at the 5.0×10^{-8} significance threshold. These false discovery rates, which add up to 0.4%, suggest that our multiple testing procedure is overall conservative. FDR approach using 5% discovery rate has been suggested robust in a previous genome-wide association publication (Nelson, C.P., et.al., Nature Genetics volume 49, pages 1385–1391 (2017)). We have added a statement describing this to the text of the manuscript:

p. 21 “We evaluated false discovery rate, assessed with the qvalue package in R. The P value cutoff of 5.0×10^{-8} corresponded to q-values of 0.0013 for the additive model and 0.0025 for the recessive model, which add up to 0.4%.”

Reviewer #3 (Remarks to the Author):

Age related hearing impairment (ARHI) and tinnitus are serious and disabling conditions which are both heritable. This manuscript describes work using large datasets to perform the largest study to date, which advances our understanding of the genetic mechanisms underlying both conditions. The study has used two samples from deCODE, Iceland which have been phenotyped using pure tone audiometry in air (PTA), the gold standard method of detecting hearing loss, and whole genome sequenced. The DHS subsample was enriched for carriers of rare mutations (4k cases with pure tone average >25dB and 7k controls) while the NIHSI subsample is highly skewed towards ARHI so cases (10k PTA >25dB) were matched with non-phenotyped population controls (almost 300k). The UKB samples, on the other hand, used self-reported hearing loss in a population sample with inadequate hearing tests and a mixture of imputed GWAS and whole exome sequencing. Two meta-analyses were performed using the same overall case control mix but different genotyping approaches. The results reveal a number of variants which have already been identified using the UKB dataset but importantly find 22 novel variants, 14 of which are rare and 6 are novel associations with hearing loss. In addition, the authors found a rare missense variant – a tandem duplication – having MAF 2% which confers an odds ratio of 4.2, which they included in a genetic risk score. The paper discusses the differences between the Icelandic and UK findings including the different genetic structures and is very well written. The manuscript is clear in its methods and the presentation of the results and appropriately adjusted for multiple testing. The findings of additional variants through application of the recessive model is of interest. There is a great deal of work here and a lot of results of importance to the hearing field in general. Consideration should be given to submitting the tinnitus/ARHI in a separate manuscript.

Major comments

1. The biggest challenge in a study such as this is to convince the reader that a study combining PTA in a genetically isolated population with a self-reported hearing loss measure in a heterogeneous population produces variants that are associated with ARHI. Here the definition of ARHI is problematic, and the study assumes that hearing loss is age-related when we know that the Icelandic samples are mixed in origin as far as deafness goes. So while the associations reported may be real, they beg the question of what phenotype the association is actually with. For those variants in both countries' datasets this is relatively straightforward but this challenge touches on the issue of the heterogeneity of ARHI. There is a no limitation paragraph to the discussion – please add one.

Response: We have now added more details to the limitation paragraph:

p.14-15 “A limitation of this work is that in the three datasets, the ARHI phenotype is defined in different ways. Because we did not restrict the definition of ARHI in terms of age of onset or severity, we performed follow-up analysis for all rare variants to make sure that the reported variants really associate with ARHI and not prelingual or childhood-onset deafness. Our results for common variants were mainly driven by the UK biobank dataset but 38 out of 52 variants replicated in the combined Icelandic datasets. The lack of replication in the DHS dataset is most likely due to smaller sample size, while in the NIHSI dataset it might be due to differences in the phenotype ascertainment, where patients are referred to NIHSI for hearing problems. Using population controls in the NIHSI dataset that have not been specifically screened for hearing impairment, will also misclassify some cases as controls. However, the effect sizes from the UK are highly correlated with effect sizes from Iceland and have consistent direction of effects. The age of onset, severity and progression of ARHI is highly variable between individuals, and future GWAS could further analyze subtypes of ARHI. The Icelandic datasets provide more details regarding these factors as well as the measures of hearing at specific

frequencies. Some ARHI variants have stronger effects on particular frequencies, while most affect all frequencies similarly.”

2. *Leading on from this, the Discussion should include consideration of potential for subtyping ARHI in future.*

Response: We have now added this sentence in the Discussion:

“The age of onset, severity and progression of ARHI is highly variable between individuals, and future GWAS could further analyze subtypes of ARHI.”

3. While Figure 1 is a clear flow diagram of the approach taken, the meta-analyses I and II are not described clearly in the Methods section. Please provide details.

Response: We have now added this paragraph to Methods:

p.20 “Because the UKB GWAS was performed on two sets of genotypes, we performed two separate meta-analyses. Both meta-analyses combined results from three GWAS using DHS, NIHSI and UKB datasets. In meta-analysis I, we used the UKB GWAS results based on the variants from the HRC reference panel and in meta-analysis II we used the UKB GWAS results based on the variants identified through WES.”

4. *The association with tinnitus – also a highly important and disabling phenotype – appears to be a bolt-on. While the subject is covered in the Introduction it doesn’t get a mention in the abstract and the data are hidden in Supplementary Table 11. Shared genetic variants between the two phenotypes are identified (n=13) that ARHI risk alleles increase risk of tinnitus. Please give suitable prominence throughout all sections.*

Response: We display the data in Figure 4 and the data behind the points in that figure are in Supplementary Table 11. We have now added more detail in the results section about the association with tinnitus:

p.11 „ARHI variants detected under the additive model were tested for tinnitus using the additive model and ARHI variants detected under the recessive model were tested for tinnitus using the recessive model. Thirteen ARHI variants associate with tinnitus, controlling the false discovery rate at 0.05 using the Benjamini-Hochberg procedure (Figure 6, Supplementary Table 11). Variants in *CTBP2*, *CRIP3*, *AGO2*, *PHLDB1*, *LMX1A*, *SLC26A5*, *ACADVL*, *SYNJ2* and *CLRN2* associated with tinnitus under the additive model and variants in *ILDR1*, *ABCC10*, *SH2D4B* and *C10orf90* associated with tinnitus under the recessive model. For all of the thirteen variants, the ARHI risk increasing allele increases the risk of tinnitus, and the effect of all the ARHI variants on ARHI risk and tinnitus risk are highly correlated ($r = 0.72$, $P = 6.2 \times 10^{-8}$ and $r = 0.86$, $P = 6.0 \times 10^{-4}$ for the additive and recessive model respectively, Figure 6).“

We also added this sentence to the abstract:

„Furthermore, we found that thirteen ARHI variants also associate with tinnitus, and the effects of ARHI variants on ARHI and tinnitus are highly correlated, suggesting that these phenotypes share genetic causes.“

Regarding the remark from the reviewer that “consideration should be given to submitting the tinnitus/ARHI in a separate manuscript”. We performed a meta-analysis GWAS for tinnitus and did not detect any genome-wide significant variants. We therefore do not think that we have enough interesting findings for a separate tinnitus manuscript.

5. Noise exposure is an important consideration and the DHS dataset is very useful in showing the influence of occupation to the risk of ARHI. However if the top GRS decile shows no increase risk with noise (P13 line 1) how do the authors justify suggesting the use of GRS as a screening tool and recommend noise avoidance (p14 line 1)?

Response: High genetic risk and noise exposure are two risk factors for ARHI. No interaction between these risk factors means that the effect of noise exposure is the same within different strata defined by the GRS. We observed that among individuals in the top GRS decile, noise exposure does associate with increased risk of ARHI similarly as for the rest of the population. Although the two risk factors do not show an interaction, the two risk factors associate with ARHI in an additive manner and individuals in both risk groups are at a higher risk of developing ARHI than those that are only in either one of the two risk groups. We can therefore conclude that avoiding noise exposure is even more important for those who are already at a high risk due to genetics.

6. There is further inconsistency around noise exposure in Discussion paragraph 3. If only 22% of people with mild ARHI report hearing handicap then again genetic screening makes no sense – hearing screening is what is indicated to demonstrate to an individual that there is actual hearing loss present not an abstract risk of it. Please amend

Response:

In general, identifying individuals at a high risk of diseases can enable enhanced screening or preventive therapies (Khera et al.). Recently, a lot of discussion has been about including GRS in clinical care. Since ARHI is often not detected until it has had many negative consequences, there is a need for better screening strategies to detect hearing impairment earlier (McMahon et al.). Because of this, it would be beneficial to prioritize those who have a high genetic risk of ARHI in routine hearing tests.

Minor comments:-

1. The abstract is a poor reflection of the totality of the study, with methods lacking. Please include the phenotyping approach used in the various cohorts.

Response: According to the submission guidelines from Communications Biology the abstract should be fewer than 150 words and include the background and context of the work and then the major results and conclusions of the paper in the present tense. There is no space for methods in such a short abstract.

2. Please make clear in Methods the difference between meta-analysis I and II

Response: See response to comment number 3 above in major comments.

3. Please add more detail around the patients included in the NIHSI dataset – are these people referred with hearing and speech difficulties?

Response: People with hearing and speech difficulties are referred to NIHSI. We added this information to the main text:

p. 4 „The NIHSI is a clinic where patients are referred to for hearing and speech difficulties, and the NIHSI dataset consists of 36,905 audiometric measures of 22,212 Icelanders (55.5% men; mean age = 48.0, SD = 32.4), of which 43.7% were performed on children (<18 years old).“

4. Please add the tinnitus methods and results to abstract

Response: We added the sentence below to the abstract.

p. 2 „Furthermore, we found that thirteen ARHI variants also associate with tinnitus, and the effects of ARHI variants on ARHI and tinnitus are highly correlated, suggesting that these phenotypes have shared genetic causes.“

5. Typo – page 7 line 3 MAF<10.0%

Response: There is not a typo in this line. If the MAF is 10%, then the expected genotype frequency in homozygous state is $0.1^2=0.01$, or 1%. Therefore, we define rare variants in homozygous state as variants with $MAF<10.0\%$.

- ReferencesAngeli, Simon, et al. "Genetics of Hearing and Deafness." *Anatomical Record*, vol. 295, no. 11, 2012, pp. 1812–29, doi:10.1002/ar.22579.
- Barrenäs, Marie Louise, et al. "The Association between Short Stature and Sensorineural Hearing Loss." *Hearing Research*, vol. 205, no. 1–2, 2005, pp. 123–30, doi:10.1016/j.heares.2005.03.019.
- Cacheiro, Pilar, et al. "New Models for Human Disease from the International Mouse Phenotyping Consortium." *Mammalian Genome*, vol. 30, no. 5–6, 2019, pp. 143–50, doi:10.1007/s00335-019-09804-5.
- Khera, Amit V., et al. "Genome-Wide Polygenic Scores for Common Diseases Identify Individuals with Risk Equivalent to Monogenic Mutations." *Nature Genetics*, vol. 50, no. 9, 2018, pp. 1219–24, doi:10.1038/s41588-018-0183-z.
- Makishima, Tomoko, et al. "Targeted Disruption of Mouse Coch Provides Functional Evidence That DFNA9 Hearing Loss Is Not a COCH Haploinsufficiency Disorder." *Human Genetics*, vol. 118, no. 1, 2005, pp. 29–34, doi:10.1007/s00439-005-0001-4.
- McMahon, Catherine M., et al. "The Need for Improved Detection and Management of Adult-Onset Hearing Loss in Australia." *International Journal of Otolaryngology*, vol. 2013, 2013, pp. 1–7, doi:10.1155/2013/308509.
- Meehan, Terrence F., et al. "Disease Model Discovery from 3,328 Gene Knockouts by the International Mouse Phenotyping Consortium." *Nature Genetics*, vol. 49, no. 8, 2017, pp. 1231–38, doi:10.1038/ng.3901.
- Mitchell, Braxton D., et al. "Using Previously Genotyped Controls in Genome-Wide Association Studies (GWAS): Application to the Stroke Genetics Network (SiGN)." *Frontiers in Genetics*, vol. 5, no. APR, 2014, doi:10.3389/fgene.2014.00095.
- Parker, Lisan L., et al. "Absence of Hearing Loss in a Mouse Model for DFNA17 and MYH9-Related Disease: The Use of Public Gene-Targeted ES Cell Resources." *Brain Research*, vol. 1091, no. 1, 2006, pp. 235–42, doi:10.1016/j.brainres.2006.03.032.
- Shen, Jun, et al. "SHIELD: An Integrative Gene Expression Database for Inner Ear Research." *Database*, vol. 2015, 2015, doi:10.1093/database/bav071.
- Sveinbjornsson, Gardar, et al. "Weighting Sequence Variants Based on Their Annotation Increases Power of Whole-Genome Association Studies." *Nature Genetics*, vol. 48, no. 3, Nature Publishing Group, 2016, pp. 314–17, doi:10.1038/ng.3507.
- Wood, Andrew R., et al. "Defining the Role of Common Variation in the Genomic and Biological Architecture of Adult Human Height." *Nature Genetics*, vol. 46, no. 11, 2014, pp. 1173–86, doi:10.1038/ng.3097.